# Over 13,000 elements from a single bonebed help elucidate disarticulation and transport of an *Edmontosaurus* thanatocoenosis

**Keith Snyder[1], Matthew McLain[2], Jared Wood[3]\*, Arthur Chadwick[3]**

**1** Biology Department, Southern Adventist University, Collegedale, TN, United States of America,
**2** Biological and Physical Sciences, The Master's University, Santa Clarita, CA, United States of America,
**3** Department of Biological Sciences, Southwestern Adventist University, Keene, TX, United States of America

\* j.wood@swau.edu

**Data Availability Statement:** Data and high resolution images for all the fossils used in this study can be found at our online repository: https://

## Abstract

Over twenty years of work on the Hanson Ranch (HR) Bonebed in the Lance Formation of eastern Wyoming has yielded over 13,000 individual elements primarily of the hadrosaurid dinosaur *Edmontosaurus annectens*. The fossil bones are found normally-graded within a fine-grained (claystone to siltstone) bed that varies from one to two meters in thickness. Almost all specimens exhibit exquisite preservation (i.e., little to no abrasion, weathering, and breakage), but they are disarticulated which, combined with our sedimentological observations, suggests that the bones were remobilized and buried after a period of initial decay and disarticulation of *Edmontosaurus* carcasses. Because of the large number of recovered fossil elements, we have been able to gain deeper insight into *Edmontosaurus* biostratigraphy including disarticulation and transport of skeletal elements. The most common postcranial elements in the bonebed are pubes, ischia, scapulae, ribs, and limb bones. By contrast, vertebrae, ilia, and chevrons are rare. The most common craniomandibular bones include dentaries, nasals, quadrates, and jugals, whereas the premaxillae, predentaries, and braincase bones are underrepresented. Thus, overall, chondrocranial and axial elements, as well as distal elements of the limbs, are distinctly underrepresented. We hypothesize that following decay and disarticulation, hydraulic winnowing removed the articulated sections (e.g., vertebral columns) and the small, distal-most elements before, or at the same time, the remaining bones were swept up in a subaqueous debris flow that generated the deposit. Comparison of the HR Bonebed with other widely dispersed Upper Cretaceous hadrosaurid-dominated bonebeds reveals many shared attributes, which suggests similar processes at work in the formation of these bonebeds across space and time. This in turn reflects similar behavior by populations of these species around the world resulting in similar modes of death, becoming interred in similar depositional settings.

fossil.swau.edu/. All other relevant data are within the manuscript.

**Funding:** The authors received no specific funding for this work.

**Competing interests:** The authors have declared that no competing interests exist.

## Introduction

*Edmontosaurus*-dominated bonebeds are widespread in the Upper Cretaceous (Campanian–Maastrichtian) strata of western North America and have been reported recently from the Dinosaur Park [1], Horseshoe Canyon [2,3], Two Medicine [4], Hell Creek [5], Prince Creek [6], and Lance Formations [7,8]. An upper Maastrichtian *Edmontosaurus annectens* monodominant bonebed of the Hell Creek Formation has recently been described by Ullmann et al. [9], with comparisons to other similar published bonebeds. By examining the host matrix, ontogenic stages, probable cause of death, state of articulation of the remains, and various bone surface modifications, a broader picture is emerging of the paleobiology and paleoecology of this species.

Limited information for the Lance Formation (upper Maastrichtian) has been presented in an unpublished thesis by Christians [10] and this Formation has been mentioned tangentially by Colson et al. (5) in a stratigraphic publication. Sternberg [11], Tedesco et al. [12], Chadwick et al. [8], and Weeks [13] have identified other *E. annectens* bonebeds in the Lance Formation, but none have been described in detail in the scientific literature.

This study examines in detail the taphonomy and depositional history of an extensive *E. annectens* bonebed in the Lance Formation of eastern Wyoming. The Hanson Ranch (HR) Bonebed includes five main quarries and three exploratory quarries. Approximately 13,000 elements (including ~8,400 identifiable bones) have been recovered in 506 m$^2$ of excavated area between 1996 and 2016. All fossils that have been collected are housed at Southwestern Adventist University (SWAU) in Keene, TX, and can be examined online at http://fossil.swau.edu. Through the description of the taphonomy of this site, and comparisons of this site with other Upper Cretaceous hadrosaurid bonebeds, we hope to gain insight into the complexity of catastrophic death assemblages and the paleobiology of *E. annectens*.

## Geological setting

The Lance Formation of northeastern Wyoming is the uppermost deposit of the Cretaceous in the Powder River Basin. It is overlain by the Paleocene Fort Union Formation and rests upon the Fox Hills Formation. The contact with the Fox Hills Formation is conformable and is often difficult to identify. Upper Fox Hills Formation sediments probably interfinger with those of the lowermost Lance Formation in most outcrops [14]. Combined with the underlying Fox Hills Formation sandstone, from which it is difficult to distinguish, the Lance Formation attains thicknesses in excess of 1 km in the southern part of the basin, as determined from well cores [14].

During deposition of the Lance, the region presently represented by the Powder River Basin was located on the western margin of the Western Interior Seaway. The southward thickening of the Lance and easterly directed paleocurrents [13,15] indicate that the tectonics responsible for forming the basin had not yet influenced sedimentation and probably did not do so in this region until well after the Cretaceous-Paleogene (K-Pg) boundary [16]. Dickenson et al. [17] recognized that marine foreland sedimentation persisted in the Powder River Basin longer than in other peripheral Laramide basins, and that in this region marine influence continued well into the Paleocene.

The Lance sediments accumulated on top of marine and marginal marine floodplains in an area of low relief [14, 18]. Paleocurrents and possible source areas for the Lance sediments indicate that the clastic materials were derived from the west, probably from west of the present Bighorn Basin [14–15, 17]. Sediments in the study area are poorly indurated, dark gray to black claystones and mudstones that weather to light gray on exposed surfaces, and tan to white, fine to medium-grained immature lithic sandstones that include granitic rock

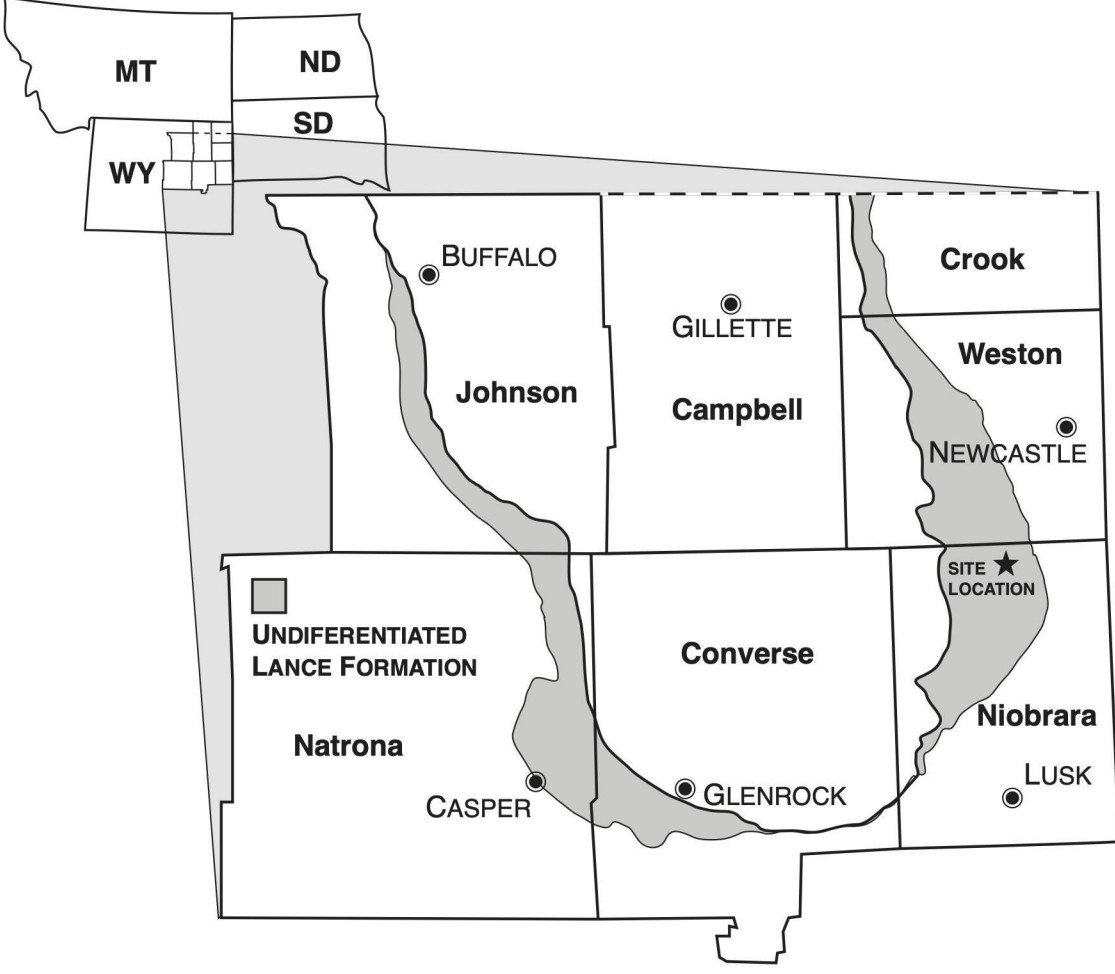

**Fig 1. Hanson Ranch Bonebed site on a map of surface outcrops of the Cretaceous Lance formation.**

fragments. Connor [14] suggested sedimentary cover of the emerging Washakie, Owl Creek, and Wind River mountains as a possible source of these sediments. Detritus derived from the emerging Granite Mountains [19] may have contributed the granitic rock fragments.

The *Edmontosaurus* bonebed that is the subject of this paper is located in the eastern limb of the Powder River syncline in north-central Niobrara County, WY (Fig 1). The bonebed occurs in the middle to upper part of the Upper Cretaceous (Maastrichtian) Lance Formation. Because of the low angle of dip ($1-3^0$ to the west), the thickness of the bone-bearing unit, and the scarcity of identifiable stratigraphic marker beds, the position of the bonebed within the Lance was estimated geometrically [20].

## Sedimentology

The Lance Formation, in the study area and generally throughout its exposed extent, consists of poorly defined strata of claystone, mudstone, and immature, fine-grained quartz and lithic sandstone. Because cementation is generally absent or is secondary calcite, much of the exposure of the Lance manifests itself as "badlands topography." Occasional beds of white sandstone are up to 5 m thick. Rapid deposition of the sandstone is attested to by beds displaying seismic disruptions that are uniform throughout the bed, but do not extend into the

underlying or overlying sediments (Fig 2), indicating that dewatering had not occurred prior to the seismic event [13].

Lance stratigraphy is problematic since marker beds are very difficult to find. The most prominent "beds" are typically poorly cemented sandstones that have secondarily been cemented with calcite. Such cementation may persist for many meters or may disappear in just a few meters. Even a persistent bed that can be traced laterally for 100 m or more, may change character completely in only 20 m, grading into fine-grained sandstones, mudstone, or even claystones. Because of this lack of persistent bedding, placing quarries in the Lance Formation in a stratigraphic framework is exceedingly difficult. Often the bones themselves provide the only reliable stratigraphic marker. Recently, a disturbed sandstone bed has been mapped over an area of approximately 50 km$^2$ [13]. Using this bed as a chronostratigraphic horizon, we have been able to place the quarries into approximate stratigraphic position by measuring downward from the seismite to an arbitrary point below the bonebed (Fig 3).

The bonebed in the main quarry area is overlain by a bed of tan, poorly cemented, fine-grained, immature quartz sandstone in conformable contact with the underlying mudstone. The sediment containing the bones is a 1–2 m thick, light to medium gray (sometimes almost black in fresh exposures), claystone to siltstone, which is also poorly to moderately indurated. Weathered surfaces, when wet, quickly revert to mud. The bones are found within the claystone, generally in the lower part, in a normally graded bed. The largest bones may rest on the bottom of the bed, or where the bed is thicker, bones appear to be entirely suspended within the mudstone. The underlying bed is often a layer of greenish, moderately well-cemented quartz, fine-grained quartz sandstone with secondary chlorite cement. This layer, generally only a few centimeters thick, may rest on an orange-weathering, fine-grained well-cemented quartz sandstone with carbonate cement.

## Methods and materials

### Specimen preparation and taphonomic classifications

The HR *Edmontosaurus* Bonebed was excavated from 1995–1999 with the removal of 135 elements. Concentrated work began in 2000 and the bed has been worked each summer since. The data in this paper extend through the 2016 field season. All specimens have been cataloged except for small *Edmontosaurus* teeth less than 1 cm, short ossified tendons less than 5 cm, and unrecognizable fragments less than 1 cm in greatest length.

The quarries were initially opened where bones were exposed at the surface. Elements were excavated using standard techniques [21]. Field techniques using pre-printed cards, digital photographs of specimens *in situ*, descriptive drawings and measurements in a field book, and high-precision RTK GPS positioning were followed as described in Chadwick et al. [22]. Data were entered into ArcGIS to create three-dimensional maps of the quarries and bones [23].

Cleaning and stabilization were completed in the laboratory. All elements were then re-photographed and their data reconfirmed and entered into the online database. Each element was photographed in 360° to allow virtual examination via the internet (see http://fossil.swau. edu). Standard taphonomic procedures were followed as developed by Behrensmeyer [24], enhanced by Ryan et al. [25] and Eberth et al. [26], and advocated for comparative purposes between bonebeds by Lyman [27] and Moore [28]. Data were usually collected for fracturing, abrasion, weathering, pathologies, surface markings, crushing, sizes, and percentage completeness. These data and information on taxonomy, anatomy, and spatial position in the quarries were used to complete each specimen profile.

We have employed the suggestions of Ryan et al. [25] as a basis for the four stages of weathering as it adds slightly to Fiorillo's [29] simplification of Behrensmeyer's [30] original

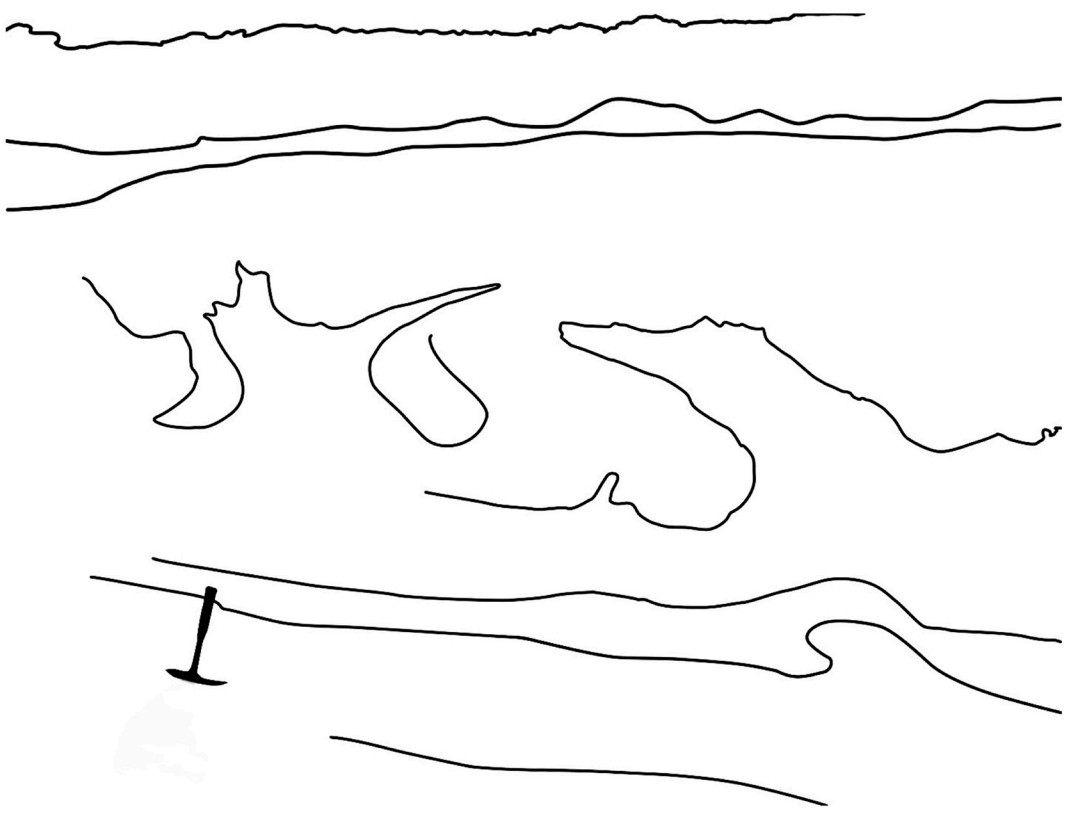

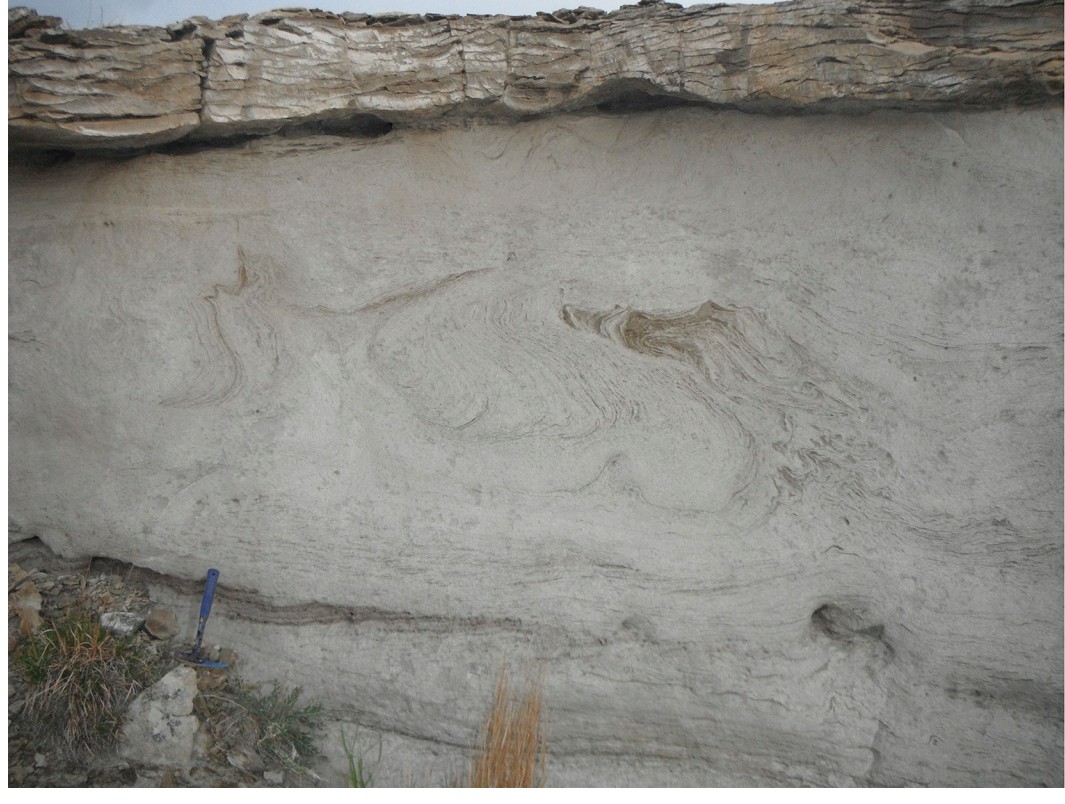

**Fig 2. Sandstone unit showing effects of seismic modification of primary sedimentary structures.** Note that the underlying beds are not affected. The capping rock is also sandstone, but it is diagenetically cemented with carbonate. Hammer bottom left for scale.

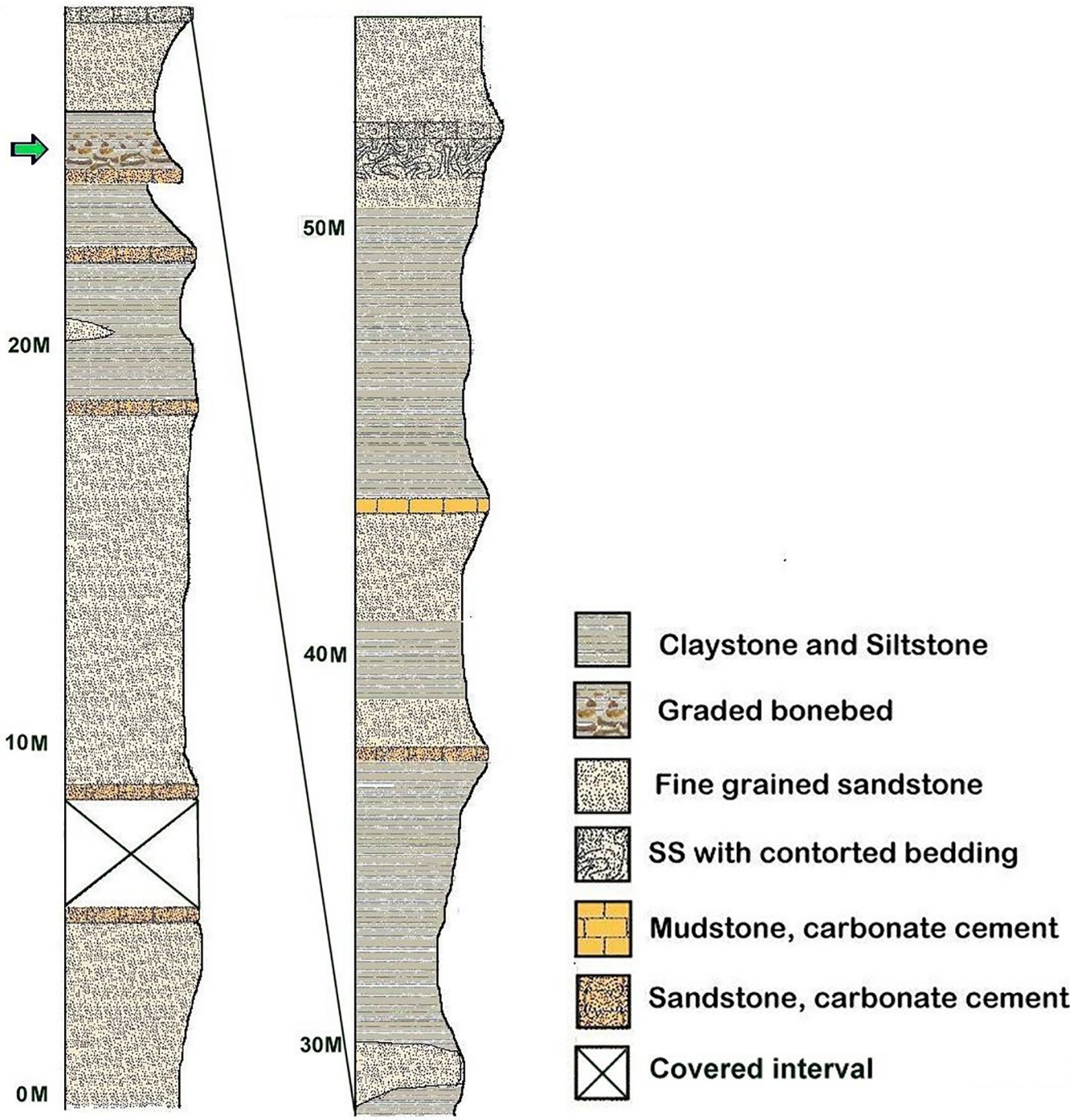

**Fig 3. Local stratigraphy of the main bonebeds at the hanson research station.** Units in meters above arbitrary base. Green arrow designates position of main bonebed in section.

publication. These stages are: Stage 0 = Fossil surface shows no sign of cracking or flaking due to weathering, Stage 1 = Surface cracking parallel or subparallel to fiber structure, Stage 2 = Cracks starting to penetrate into the marrow cavity and surface flaking away from cracks, Stage 3 = Large chunks of outer laminated bone have flaked away.

The four abrasion categories classified by Ryan et al. [25] were also used. These stages are: Stage 0 = Pristine and unabraded surface, Stage 1 = Broken edges are rounded, Stage 2 = All edges are well-rounded, and Stage 3 = All edges extremely rounded, processes appear as bone protrusions only. The three major types of bone fracturing used for classification (longitudinal, transverse, and spiral/greenstick) were originally from Haynes [31] and subsequently clarified by Ryan et al. [25]. We have added "indeterminate" as an additional classification for those not readily fitting into these categories.

Element orientation was obtained from the database, by isolating 268 femora, tibiae, fibulae, humeri, radii, and fibulae from the four largest quarries. Orientation was analyzed in R (version 3.6.0) using *CircStats* (version 0.2–6) and *Circular* (version 0.4–93) packages. Rose diagrams were also generated in R using the *ggplot2* package (version 3.1.1). Spearman's correlation analyses were done using R and the *stats* package (version 3.6.0).

Due to the continued challenges of using Voorhies groupings for element transport and distribution, we followed the suggestion of Gangloff and Fiorillo [6] to instead consider the number of individual elements from the HR Bonebed to give a better understanding of the possible survivability and distribution of elements.

Approximately 11,000 citizen scientist working days were recorded, enabling public participation by well over a thousand people over 20 years of excavation.

## Species identification

In the northwestern U.S. and southwestern Canada, two species of *Edmontosaurus* are recognized: *E. regalis* (naming citation) and *E. annectens* (naming citation). The main quarries expose a monodominant bonebed of this genus. Campione and Evans [32] assert that *E. regalis* are found in the upper Campanian and *E. annectens* are found in the upper Maastrichtian. *E. regalis* appears to be limited to the upper Campanian and lower Maastrichtian [33]. Ullmann et al. [9] agree and have built a strong case for Hell Creek material belonging to *E. annectens*. This species is found in the Lance, Hell Creek, Laramie, Scholard, and Frenchman formations of Colorado, Wyoming, South Dakota, Montana and Alberta, all of which pertain to the upper Maastrichtian [34]. The HR bonebed is located in the southern portion of the range of the Lance Formation, leading us to conclude all hadrosaurine elements are of the species *E. annectens*.

In addition to identifying *E. annectens* based on biostratigraphic and geographic occurrence, we used several diagnostic features described by Bell and Campione [32] to distinguish between *E. annectens* and *E. regalis*. Bell and Campione [32] describe *E. regalis* as having strongly excavated posterodorsal margins on the narial vestibule of the nasal bone, a ventral expansion at the rostral tip of the anterodorsal process, a strongly developed postorbital fossa and laterally expanded jugal process of the postorbital, a premaxilla possessing a posteriorly expanded oral margin, and an expanded dorsal jugal process. Using these features, we comparatively analyzed eight nasal elements, 13 postorbital elements, and a single premaxilla and jugal. All eight nasal elements exhibited weakly excavated posterodorsal margins on the narial vestibules and lacked a ventral expansion on the rostral tip of the anterodorsal process. The postorbital fossae were weakly developed in all 13 elements we analyzed, and all postorbital elements lacked a laterally expanded jugal process. Furthermore, the single premaxilla lacked a posteriorly expansion, and the single dorsal process of the jugal was straight and narrow.

We also used the quantitative diagnostic features described by Xing et al. [33] for the postorbital and nasal bones in our study to distinguish between *E. annectens* and *E. regalis*. They describe the opening of the postorbital fossa as being 68% as wide as the frontal in *E. regalis*, compared to 50% for *E. annectens*. The openings the postorbital fossae in our study are all very narrow and closer to the 50% value described for *E. annectens*. Altogether, these character diagnostics strongly support *E. annectens* as the species found in the Hanson Ranch Bonebed.

## Results

### Attributes of Bonebed

The HR Bonebed is exposed along the eastern margin of a local transition from pasturelands in the east to broken topography of canyons and ridges in the west, referred to locally by the community as "The Breaks." This erosional relief has exposed an extensive layer of bones along the sides of four successive canyons arising from a single ridge. This bonebed is the subject of much of our research and at present is the target of our five main quarries: North, South, Southeast, Teague, and West in addition to three exploratory quarries: Toe, Neufeld, and DKC (Fig 4). Erosion has removed the main bonebed on the west, south, and southeast sides, limiting the ability to measure the original extent of the bonebed. The exploratory quarries are beginning to help define the northern and northeastern boundaries of this apparently contiguous layer. Areas between the quarries have as much as seven meters of overburden, limiting access in places.

Typically, the bones are found in mudstone to claystone, which is easily removed from the specimen. However, some bones are partially or entirely encased in carbonate-cemented mudstone that is very hard and generally leaves the bones affected by the removal process. The formation of a carbonate case is thought to result from the release of organic constituents, particularly fatty acids, to the surrounding matrix [35,36]. This and the production of ammonia from protein degradation probably caused the precipitation of carbonate in the matrix.

### Bone density

Our five main quarries, examining the thickest portion of the bonebed, include North (N), South (S), Southeast (SE), West (W), and Teague (T). Three exploratory quarries were also opened to determine the lateral extent of the bonebed; these include Toe, Neufeld (Neu), and DKC. The total excavation of 508 m$^2$ has yielded over 13,000 specimens including identifiable

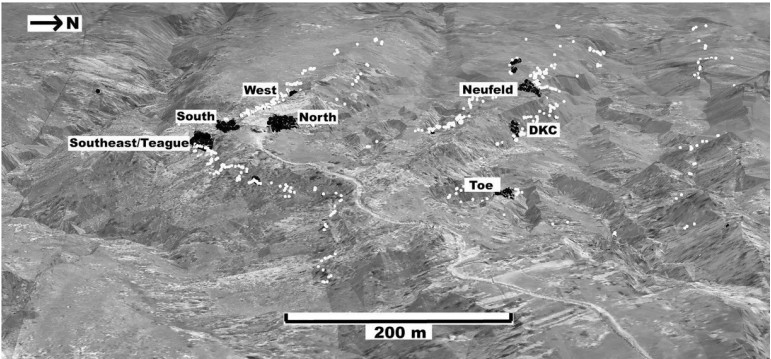

**Fig 4. Three-dimensional aerial view of the hanson ranch field area including main and exploratory quarries discusses in the text.** White dots indicate location of individual bones, outlining the bonebed layer through the hills between formal excavation sites. The black dots indicate where bones were collected in the quarries.

**Table 1. Bone densities (e/m$^2$) in the five main (North (N), South (S), Southeast (SE), Teague (T), West (W)) and three exploratory quarries (Toe (T), neufeld (Neu), DKC).**

| Quarries | N | S | SE | T | W | Toe | Neu | DKC | Total |
|---|---|---|---|---|---|---|---|---|---|
| Square meters excavated | 190 | 129 | 78 | 45 | 20 | 20 | 20 | 6 | 508 |
| Total bones | 5385 | 3858 | 2318 | 1053 | 554 | 123 | 304 | 30 | 13,625 |
| Bone density (e/m$^2$) | 28.3 | 29.9 | 29.7 | 23.4 | 28 | 6.2 | 15.2 | 5 | 26.8 |

bones, teeth, ossified tendons, and bone fragments (Table 1). Screen washing of approximately 0.5 m$^3$ was implemented in several of the quarries. Specimens recovered from these efforts are included in Table 2.

Table 1 summarizes the square meters excavated, the total bones removed, and the bone density per square meter in the eight quarries. Although small by excavated area, the thickness of the bone layer, and skeletal element density, support West quarry as part of the main bonebed. Density numbers of approximately 23–30 elements per square meter (e/m$^2$) are similar to those reported for a *Centrosaurus* bonebed in the Dinosaur Park Formation [25], suggesting that these two beds may share a similar taphonomic history. The three exploratory quarries (Toe, Neufeld, DKC) have fewer elements per square meter, but vary in element concentration. Although Neufeld quarry (15 e/m$^2$) is farther away from the main quarries than either Toe quarry (6 e/m$^2$) or DKC quarry (5 e/m$^2$), it has a higher concentration of elements per square meter. Further quarrying may elucidate the reason for this difference.

A representative quarry map for North quarry is shown in Fig 5. The thicknesses of the HR bonebed in the main quarries of 90–140 cm and 30–50 cm in the exploratory quarries fall within recently reported dinosaur bonebed thicknesses from other Cretaceous sites across North America: 35–90 cm [6], 40–60 cm [38], 40–150 cm [39], and maximum thicknesses for three sites of 60 cm, 120 cm, and 150 cm [3]. That the bonebed horizon is thicker in the main quarries and thinner in the outer, exploratory quarries is consistent with a lenticular geometry for this layer.

Of the total elements recovered in these quarries (13,625), approximately 14.7% are unidentifiable bone fragments, 6.7% are *Edmontosaurus* teeth, and 17.2% are tendons (Fig 6).

## Articulation

In the HR Bonebed, skeletal elements are completely disarticulated with three exceptions. In North quarry, three complete *Edmontosaurus* caudal vertebrae were found articulated as were a femur, tibia, fibula, calcaneum and astragalus. The only other associated bones were two *Edmontosaurus* tibiae found adjacent to each other in South quarry that appeared to be a paired set. Thus <0.1% of the bones found show actual or possible articulation.

## Vertical distribution

All quarries exhibit fairly uniform normal grading of the bones, with smaller bones at the top and larger bones at the bottom. The first bones encountered are consistently unguals or phalanges, then smaller ribs, vertebrae, indeterminate bone fragments and chevrons. Large bones such as tibiae, fibulae, and femora and metatarsals are consistently found at the very bottom (Fig 7). Southeast and Teague quarries are more condensed in the lowest layer, resulting in many bones in contact. Ossified tendons and teeth are found randomly throughout the bonebed.

**Table 2. HR Bonebed taxa represented.** *Nanotyrannus* designation based on Larson [37].

| Mollusca | | | | | Number of Elements |
|---|---|---|---|---|---|
| | Gastropoda indet. | | | | 27 |
| | Pelecypoda (Bivalvia) indet. | | | | 16 |
| Chondrichthyes | | | | | |
| | Rajiformes–Rajidae (skates) | | | | 3 |
| Actinopterygii | | | | | |
| | Lepisosteiformes–Lepisosteidae–*Lepisosteus* (gar) | | | | 5 |
| | Actinopterygii indet. | | | | 18 |
| | | Teleostei indet. | | | 2 |
| Reptilia | | | | | |
| | Reptilia indet. | | | | 165 |
| | Testudines indet. (turtles) | | | | 74 |
| | Squamata indet. (lizards and snakes) | | | | 4 |
| | Archosauria | | | | |
| | | Aves indet. (birds) | | | 30 |
| | | Crocodylia indet. | | | 34 |
| | | | Crocodyloidea—*Leidyosuchus* | | 17 |
| | | | Alligatoroidea—*Brachychampsa* | | 34 |
| | | Saurischia | | | |
| | | | Theropoda–indet. | | 121 |
| | | | Tyrannosauridae | | |
| | | | | *Tyrannosaurus* | 56 |
| | | | | *Nanotyrannus* | 135 |
| | | | | *Aublysodon*–like forms | 3 |
| | | | Troodontidae—*Troodon* | | 52 |
| | | | Dromaeosauridae | | |
| | | | | *Acheroraptor* | 128 |
| | | | | *Dromaeosaurus* indet. | 3 |
| | | | Coelurosauria indet.—*Richardoestesia* | | 5 |
| | | | Ornithomimosauria—*Struthiomimus* | | 4 |
| | | Ornithischia | | | |
| | | | Hadrosauridae–Probable *Edmontosaurus* | | 12,503 |
| | | | Pachycephalosauridae–*Pachycephalosaurus* | | 5 |
| | | | Ceratopsidae—*Triceratops* | | 103 |
| | | | Nodosauridae—*Nodosaurus* | | 4 |
| | | | Thescelosauridae—*Thescelosaurus* | | 4 |
| Mammalia indet. | | | | | 2 |

The variations in bonebed thickness appear to mirror undulations of the upper surface of the underlying sandstone. These differences are reflected in slight variations in large bone distribution. An example of this is the higher concentration of large bones in the western part of North Quarry (Fig 5).

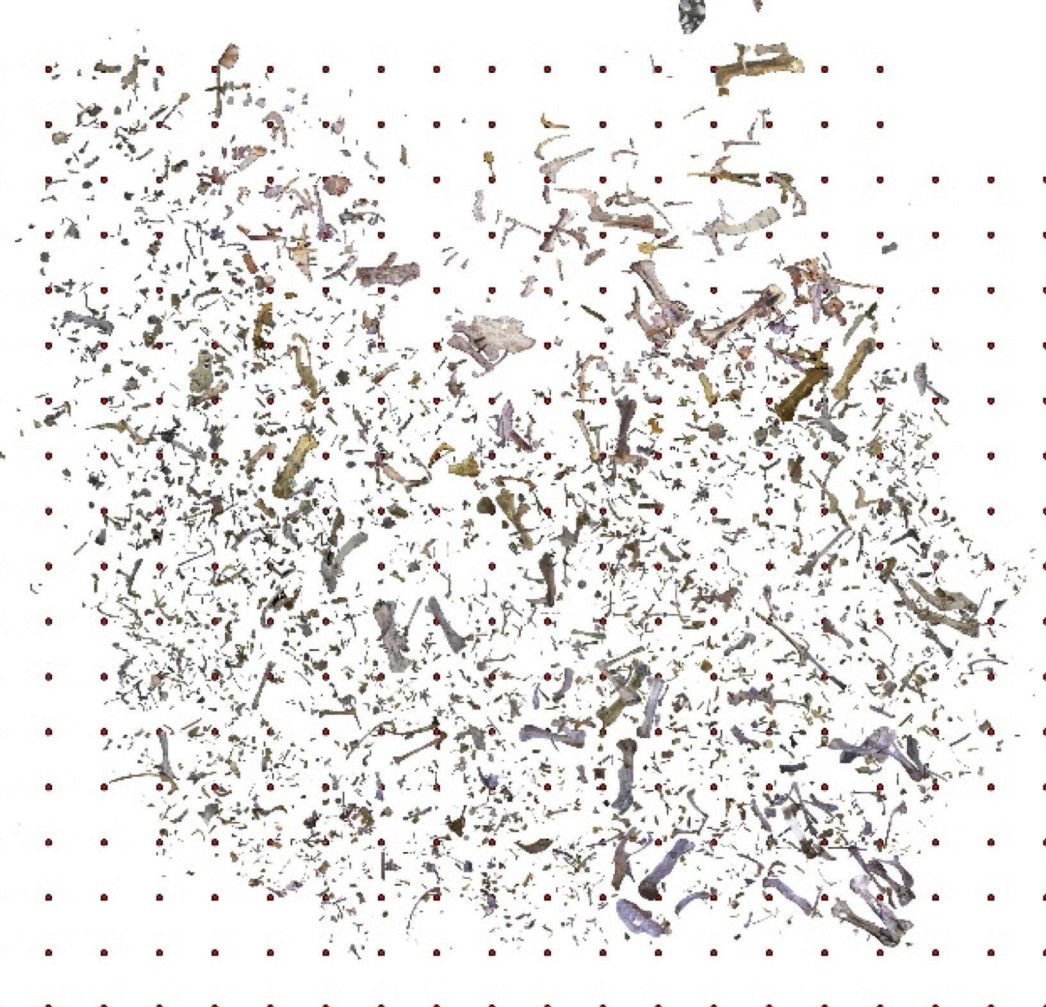

**Fig 5. Plan map of the elements excavated in a representative quarry (North).** Marker dots are spaced in meter increments. North is up.

### Element orientation

Element orientation of 268 long bones from the four main quarries shows no significant directionality (p = 0.170) based on a Rayleigh test of uniformity (Fig 8A). Separating the bones by quarry also shows no significant directionality in the three largest quarries (North, p = 0.373, Fig 8B; South, p = 0.224, Fig 8C; Southeast, p = 0.142, Fig 8D), but the fourth quarry has a definitive mean directionality of N-S (Teague, p = 0.015, Fig 8E) which is consistent with the general trend seen in the other quarries where the trend was not statistically significant. However, due to the very close proximity of these four quarries, other factor such as unevenness of the floor or small sampling size may have influenced the directionality of long bones as they were deposited. Further excavation in the quarries will hopefully clarify these possibilities. No plunge data were analyzed since most elements were found in horizontal or near-horizontal orientation.

### Taxon representation

The HR Bonebed has produced a wide range of taxonomically diverse organisms. In terms of body size, Dinosauria is by far the largest taxon represented (>95%), with *E. annectens*

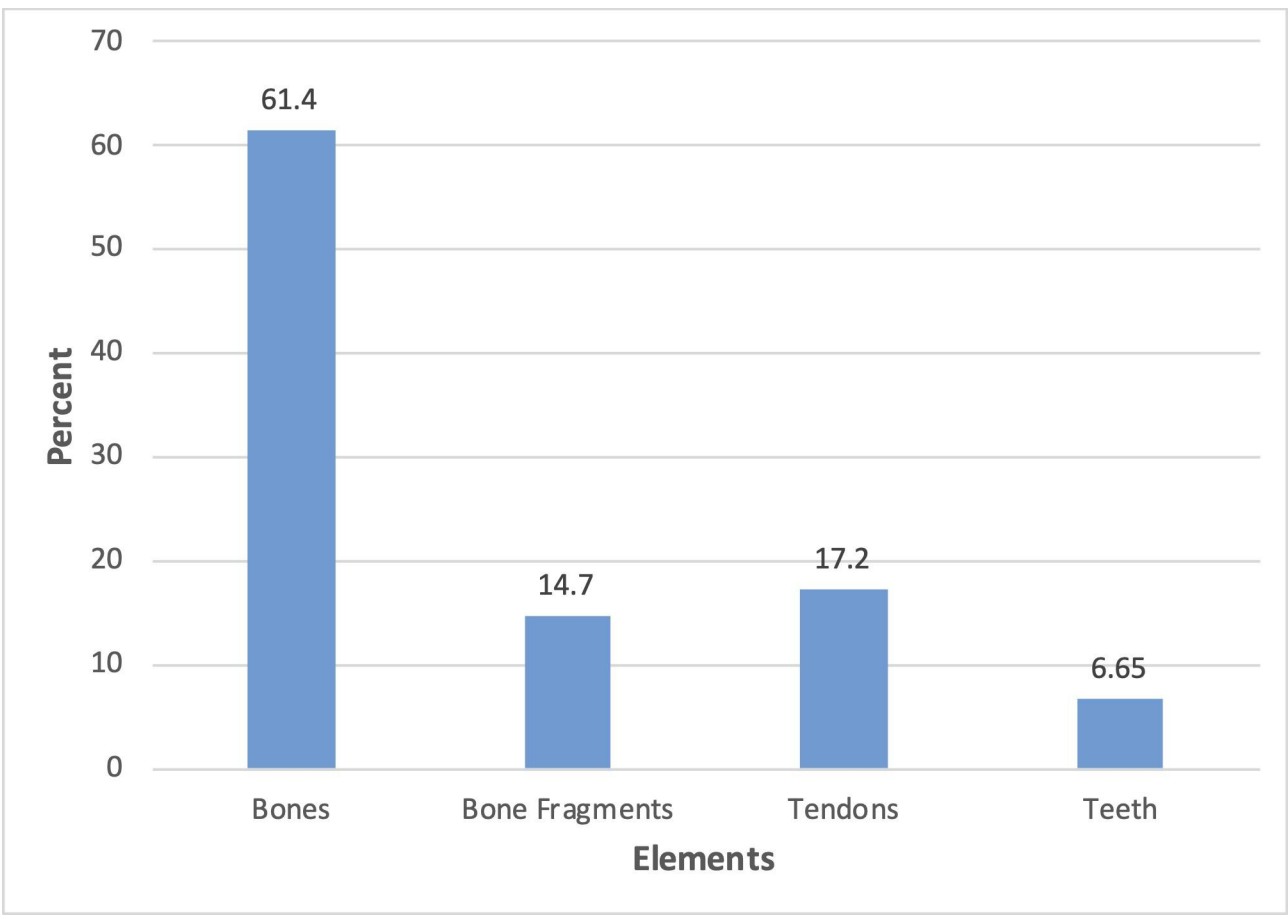

**Fig 6. Skeletal element distribution for main quarries.** Percentages out of 12,303 *Edmontosaurus* elements only.

providing the bulk of these elements, averaging about 94% in each of the four main quarries (Table 2). These counts include all elements found that are identifiable taxonomically; this filter (identifiability) eliminates about 400 elements from the total recovered. The taxa represented are consistent among the eight different quarries.

The identifiable elements include nine genera of Saurischia and five of Ornithischia. Other non-dinosaurian terrestrial taxa represented include squamates and mammals. Additionally, there are a variety of aquatic creatures, represented by crocodiles, turtles, gar and other ray-finned fish, elasmobranchs, and both gastropod and bivalve mollusks. Thus, the HR Bonebed is multitaxic and monodominant [40].

Fish are known from scales, teeth, and vertebrae, each in about equal proportions, in the HR Bonebed. Gar are the most common fish (as in the Hell Creek Standing Rock Hadrosaur Site [9]), but we have recovered only isolated teeth, vertebrae and scales of gars from the main quarries. Turtles were identified from broken carapaces and plastrons, as well as from isolated limb bones. *Leidyosuchus*, *Brachychampsa*, and perhaps other crocodylomorphs are recognized by the presence of teeth, scutes, some skull fragments and limb bones.

While the quarry is clearly dominated by the remains of Hadrosauridae, we do find some remains referable to Ceratopsidae, and a few skeletal elements assignable to Pachycephalosauridae, Nodosauridae, and Thescelosauridae dinosaurs (Table 2). Large and small theropods must have been present where the dead animals accumulated. The shed teeth of theropods are

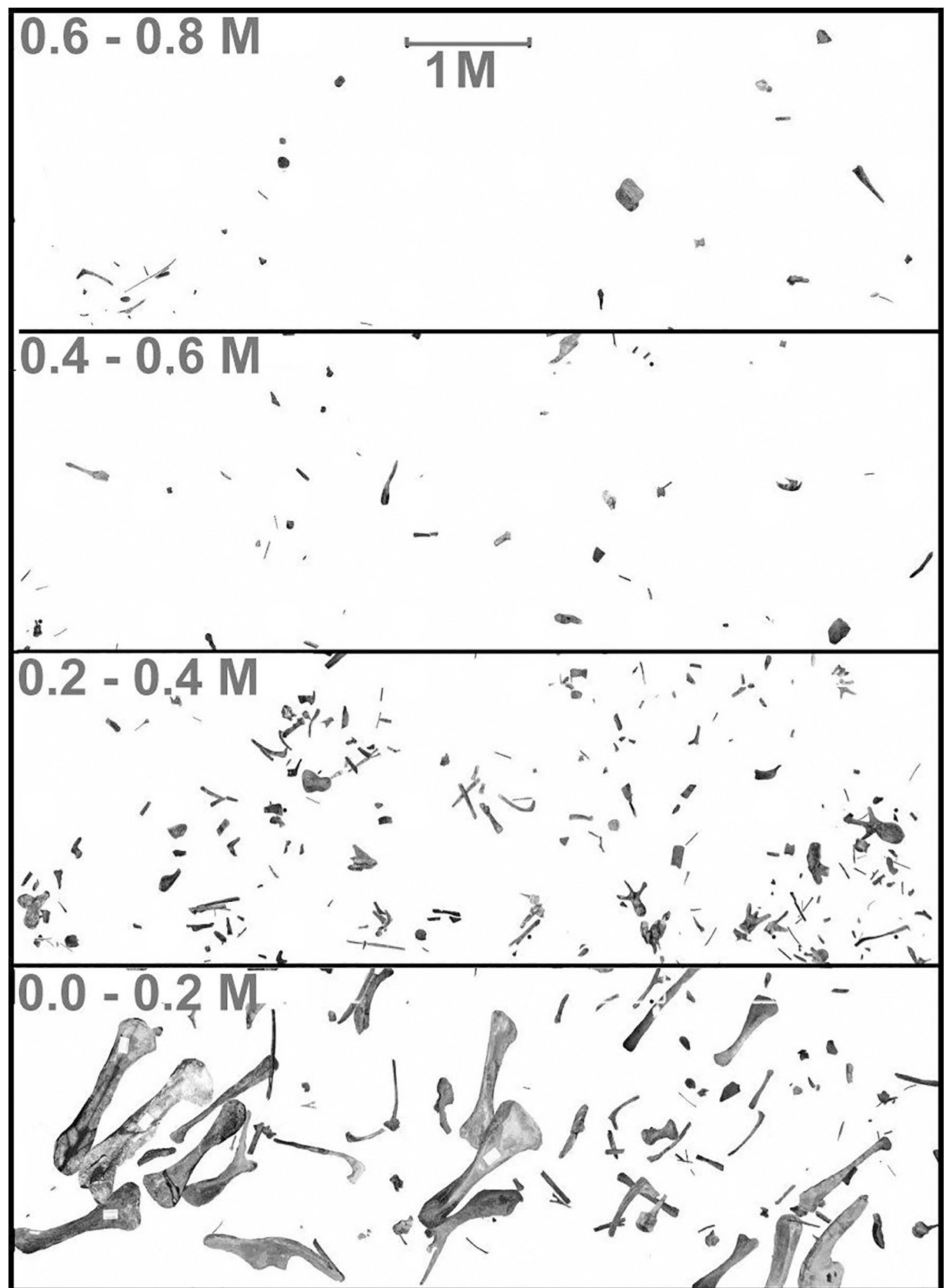

**Fig 7. Plan view of bone distribution in successive 20 cm deep intervals at north quarry.** Bar is one meter long. Stratigraphic height starts with "0.0" at the bottom.

common in the bonebed, as would be expected if they were scavenging the decaying *Edmontosaurus* carcasses. The most common of these pertain to the theropod families Tyrannosauridae, Dromaeosauridae, and Troodontidae.

In 2001, the remains of the foot of a *Nanotyrannus lancensis* were found on the surface in a nearby site designated Stair Quarry (not included in this study). Over the next 15 years, excavation of the site yielded 50 additional bones from this specimen, including a right maxilla with teeth and a left dentary with teeth. Although not yet formally described, these remains have enabled us to clearly distinguish slender, blade-like shed teeth of *Nanotyrannus* [37] from the more robust crushing teeth of *Tyrannosaurus*, both of which are commonly found in the bonebed.

Mammal fossils, identified only from teeth, are exceedingly rare in the bonebed. Several small, thin-walled hollow bones were also recovered that lacked proximal and distal ends.

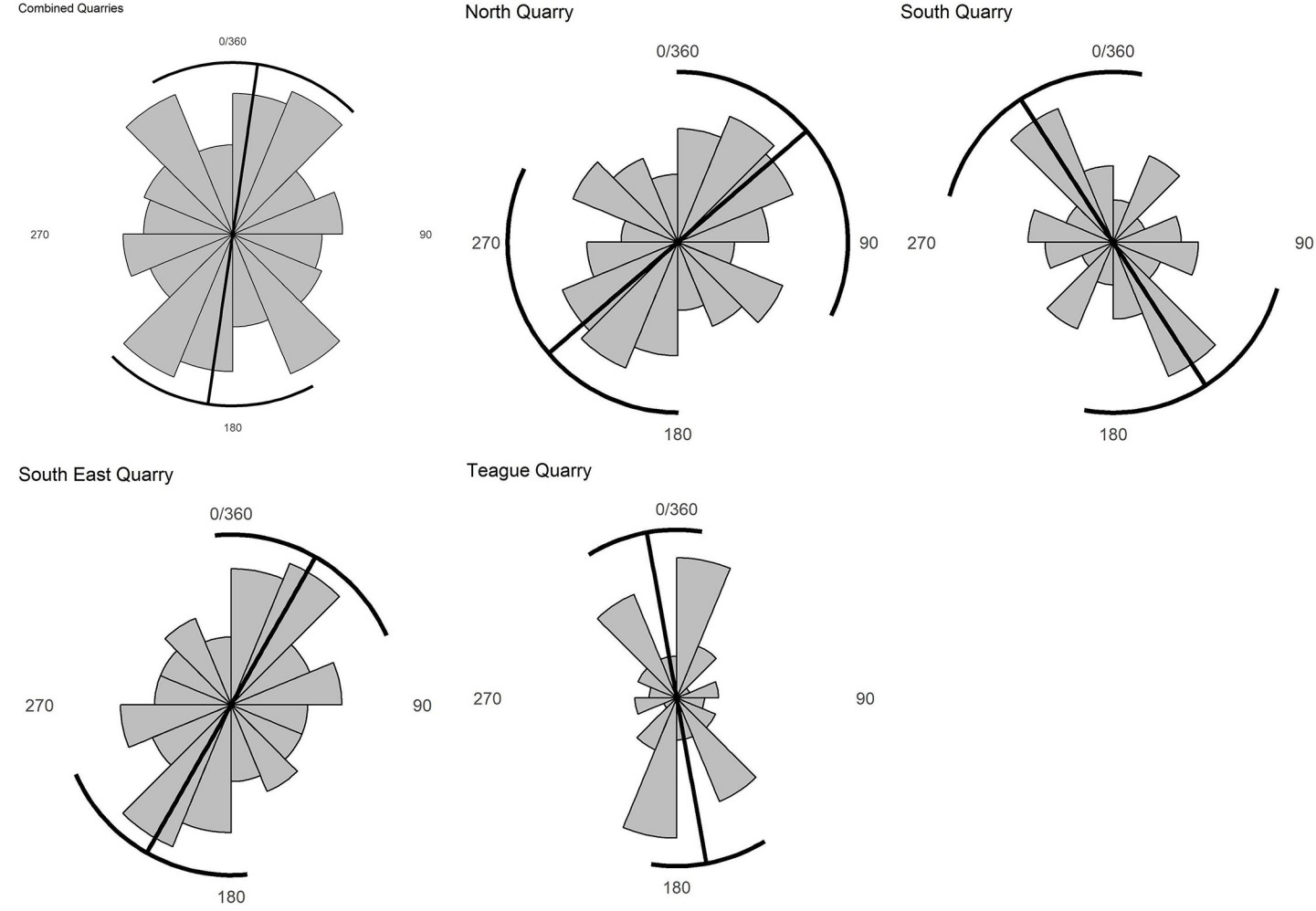

**Fig 8. Long bone orientation in the main quarries.** Outer lines are bootstrapped 95% confidence intervals.

These bones are probably from birds, a number of which have been recognized in the Lance/ Hell Creek from more complete specimens [41].

## Age distribution / ontogenic representation

Lengths of complete long bones from the HR Bonebed are displayed in Fig 9. The almost continuous size range from approximately 50% of adult length to maximum adult length correlates with late juvenile, subadult, and adult in the hadrosaurid *Maiasaura* [42]. This is similarly applied to *Edmontosaurus* [6,9] and fits quite well into the adult and subadult categories for ceratopsians used by Ryan et al. [25]. In most categories, there is a single "outlier" which is at least 5% longer than the next longest bone. We removed this outlier and averaged the next three longest bones to get a defined maximum length for mature adult long bones. Following the definitions of Ryan et al. [25], looking closely at two-thirds and half this average element length should help differentiate adult, sub-adult and juvenile elements. However, there is a continuum in bone lengths with no clear separation into discrete size groupings. There are no femora smaller than 50% the size of the largest femur in the bonebed. Tibiae, fibulae, humeri, ulnae, and radii include only one to a few elements slightly below 50% the size of the largest of each of these skeletal elements.

## Element distribution

Through the 2016 field season, we have found 8,460 identifiable E. *annectens* skeletal elements in the HR Bonebed. The most abundant skeletal elements so far relative to the expected number [1] are pubes, with a count of 114. Of these, 61 are left and nine are too incomplete to be certain which side they derive from. Thus, conservatively, 61 is the minimum number of individuals (MNI) for the assemblage (Table 3). Since there are 389 elements on average in a hadrosaurid skeleton (lambeosaurine [1]), we would expect a complete bone count of 23,729. The 8,460 identifiable elements so far recovered from the HR quarries represent a 36% recovery rate.

Percentages of both recovered postcranial and cranial elements are illustrated for a single skeleton in Figs 10 and 11, respectively. We found no major differences between the relative representation of cranial versus postcranial elements. For the postcranial appendages, numbers of recovered elements decrease in a proximal to distal progression. Pubes and ischia in the pelvic girdle and scapulae in the pectoral girdle are the most commonly recovered appendicular elements. Major long bones are next in number, followed by more distal elements.

Considering the skull (Fig 11) as an entity separate from the body, we would assume disarticulation to reach a similar extent as that of the rest of the body during decomposition. In the body, both the core and distal elements are found least frequently, whereas "intermediate elements" (scapula, humerus, ischium, pubis, femur, tibia) are found most frequently; these are also the largest skeletal elements in the body. This pattern is replicated in the skull. The small, "core" elements (of the braincase and immediately surrounding the braincase) and small elements located farthest away from the "core" of the skull (premaxilla and predentary) are found least frequently, whereas larger cranial elements (i.e., the dentary, nasal, and quadrate) are found most commonly, with intermediate-sized bones falling in between in terms of abundance.

## Weathering

Bones in each of the HR quarries are uniform in both type of bone modification and comparative percentiles within each category of modification (Table 4). Data presented here are combined data from the four largest quarries, North (N), South (S), Southeast (SE), and Teague

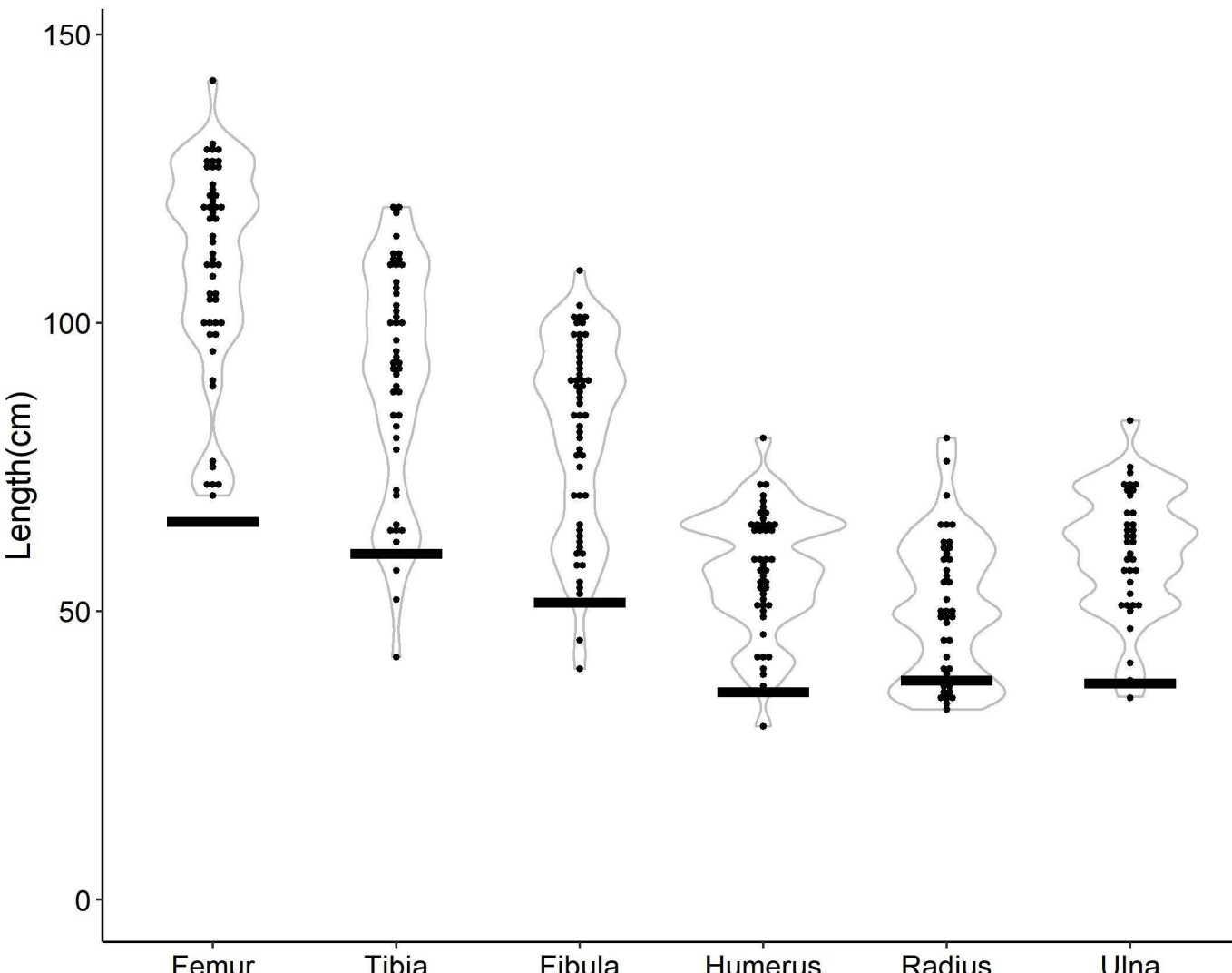

**Fig 9. Lengths of major long bones.** Dark black bars denote the size of a bone 50% the average length of the three longest specimens for each bone type (after the longest "outlier" is removed for each bone type [all categories except tibia]).

(T). The vast majority of bones show no weathering (98.5%). Data from teeth and ossified tendons were not included for weathering and abrasion because they exhibited no wear. Bone fragments are included and mirror the percentages from identifiable elements.

## Abrasion

The four stages defined by Ryan et al. [25] were applied to the 6,896 elements analyzed for abrasion. About 97 percent of these elements exhibit stage 0, showing little or no abrasion (Table 4). Those exhibiting stage 1 (with broken edges that are slightly rounded) represent about 2% of the total. Rarely do any of the bones exhibit stage 2 (0.7%) or stage 3 (0.1%) abrasion. The majority of bone fragments show little to no abrasion and mirror the percentages collected from identifiable elements. They usually exhibit well-defined edges with very little or no rounding observable.

**Table 3. Percentages of bones recovered based on MNI of 61, sorted by abundance.**

| Region | Element | % Total | | | |
|---|---|---|---|---|---|
| Cranial | | | Axial | | |
| | Nasal | 66 | | Rib | 68 |
| | Quadrate | 62 | | Rib, cervical | 10 |
| | Jugal | 52 | | Vertebra, cervical | 18 |
| | Surangular | 46 | | Vertebra, caudal | 17 |
| | Maxilla | 45 | | Vertebra, dorsal | 7 |
| | Quadratojugal | 39 | | Chevron | 15 |
| | Splenial | 38 | Pectoral Girdle and Forelimb | Scapula | 66 |
| | Hyoid | 36 | | Radius | 50 |
| | Pterygoid | 36 | | Humerus | 48 |
| | Squamosal | 26 | | Ulna | 39 |
| | Premaxilla | 24 | | Clavicle | 27 |
| | Exoccipital | 21 | | Metacarpal | 21 |
| | Angular | 19 | | Phalanx, manus | 18 |
| | Ectopterygoid | 18 | | Ungual, manus | 14 |
| | Palatine | 16 | | Coracoid | 8 |
| | Basisphenoid | 15 | | Carpal | 2 |
| | Alisphenoid | 12 | Pelvic Girdle and Hindlimb | | |
| | Parietal | 9 | | Pubis | 100 |
| | Articulated Braincases | 7 | | Ischium | 85 |
| | Predentary | 2 | | Fibula | 67 |
| | Articular | 1 | | Femur | 49 |
| | Postfrontal | 30 | | Tibia | 46 |
| | Lacrimal | 23 | | Ilium | 27 |
| | Prefrontal | 21 | | Metatarsal | 25 |
| | Frontal | 18 | | Phalanx, pes | 16 |
| | Postorbital | 14 | | Astragalus | 15 |
| | Dentary | 96 | | Calcaneum | 11 |
| | | | | Ungual, pes | 11 |
| | | | | Sacrum | 9 |

## Breakage and fracturing

We have added the "indeterminate" category to the three major categories of fracturing (longitudinal, transverse, and spiral/greenstick) used by Ryan et al. [25]. We then applied these definitions to the 8,167 bones, teeth, tendons, and bone fragments analyzed for breakage and fracturing. The most abundant fracturing category, greenstick/spiral (30.8%), arises through the breakage of bone while still fresh [43]. However, about 48% of bones collected from the HR Bonebed are unfractured (Table 4). This is in contrast to many other hadrosaur bonebeds around the world which report single digit percentages of unbroken bones (Eastern Russia [44]; Utah, USA [45]: Alberta, Canada [46]).

Less than 4% of our bones exhibit identifiable tooth marks even though we have recovered a number of theropod teeth.

## Discussion

### Element distribution and transport

The significance of element distribution in disarticulated bonebeds has been debated ever since Voohries [47] created three categories of transport for mammalian elements (primarily

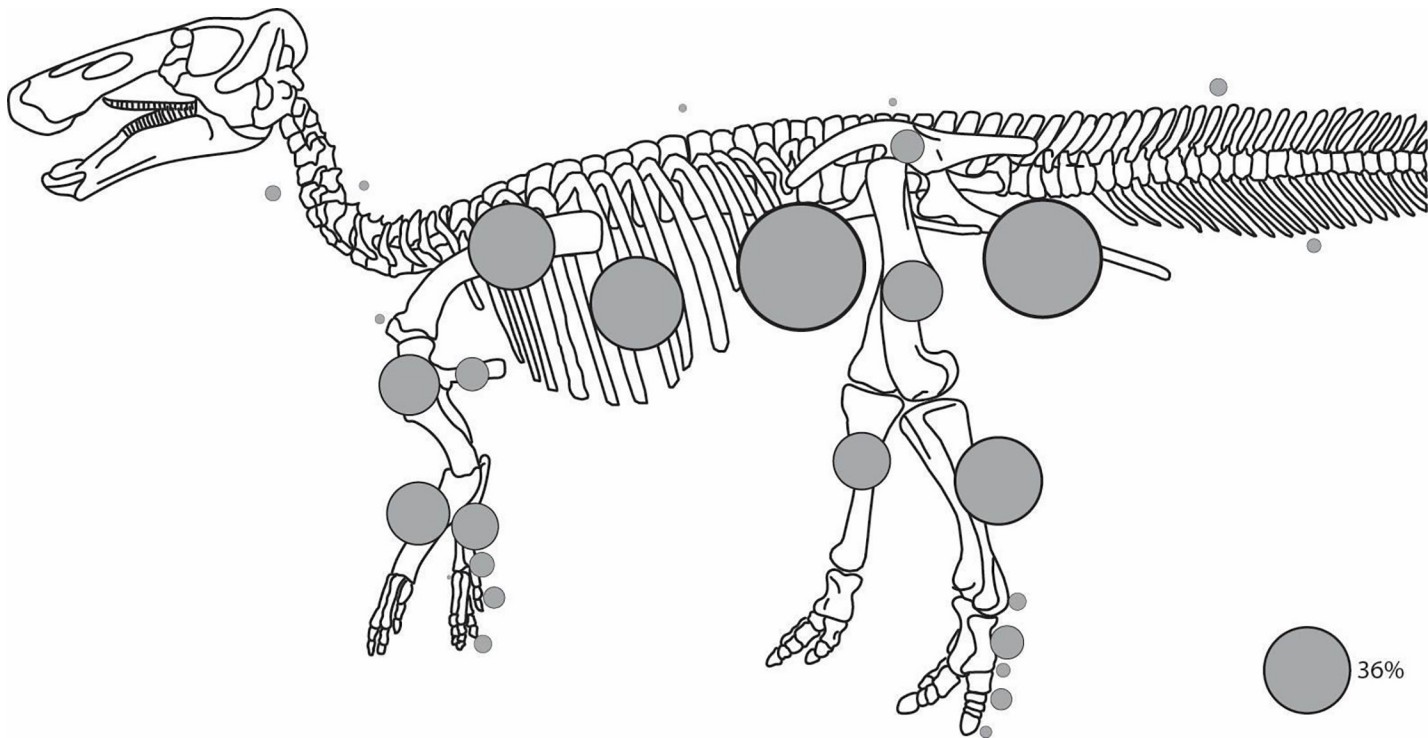

**Fig 10. Relative number of *Edmontosaurus* postcranial bones found that are over- (larger) or under- (smaller) represented compared to a 36% overall recovery rate in the bonebed.**

sheep and coyote) in 1969. His Group I bones were transported by saltation or flotation and included ribs, vertebrae, sacra, and sterna. Those in Group II were transported by traction and included femora, tibiae, humeri, pelves, radii, and metapodia. Those elements that moved very little, the skull and mandibles, he placed in Group III. Light elements were transported over larger distances than heavier ones [47]. Interestingly, scapulae, phalanges, and ulnae were lumped as intermediary between groups I and II. Thus, he fully recognized the challenges of defining few, specific categories.

Behrensmeyer [48] broadened the data available by using bones from a wider variety of animal species including hippopotamus, zebra, antelope, crocodile, and fish. She found that density appeared to be more important than size in determining whether or not bones will disperse. Shape was an important factor only for particular bones. The Voorhies groups were updated by Lehman [49], and used by Ryan et al. [25] for *Centrosaurus* and Bell and Campione [46] for *Edmontosaurus*, by adding ischia and metapodia to group I, reassigning ribs and dorsal vertebrae to group II, and reassigning sacra and scapulae (from group I), as well as femora, tibiae, and humeri (from group II), into group III.

Because Voorhies groups are based on extant animal elements and density plays such an important role in transport, and because we can never know the native densities for bones of extinct dinosaurs, the blind application of Voorhies categories is misguided. Gangloff and Fiorillo [6] did not find these categories useful for the Liscomb bonebed but attributed that to the highly fragmented nature of skeletal elements in that assemblage. Fiorillo et al. [50] found that the majority of elements they recovered were from Voorhies group II, and that heavy elements were over-represented while light elements were under-represented. This appears to be

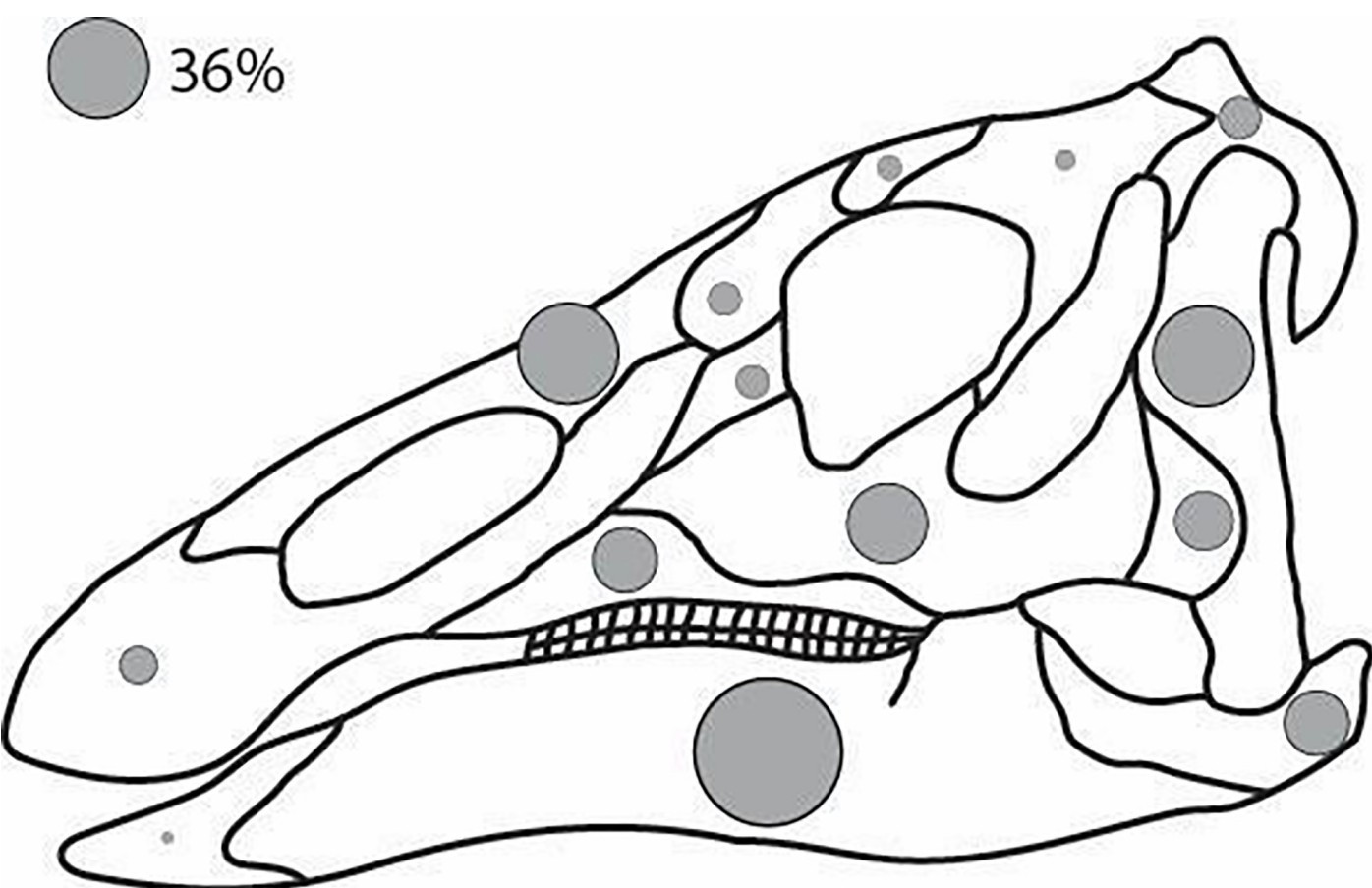

**Fig 11. Relative number of *Edmontosaurus* cranial bones found that are over- (larger) or under- (smaller) represented compared to a 36% overall recovery rate in the bonebed.**

a common finding in many quarries containing disarticulated skeletons of centrosaurines [25] and hadrosaurids [44, 46, 51, this study].

Britt et al. [45] suggested using shape (block, rod, flat, complex) as a different solution to this problem of transportation and winnowing. However, they found that this categorization was an imperfect solution since some shapes were still under-represented (block), and others were over-represented (complex) compared to expected outcomes.

The large number of elements found in the HR Bonebed helps clarify this perplexing problem. A very interesting pattern is emerging, in that core elements of the torso, vertebrae in general, and sacra in particular, are under-represented (Fig 10). Ilia are also much lower in number than expected based on the other pubic and pectoral girdle elements. This is perhaps because the ilia are more tightly bound to the sacrum than they are to either the pubis or ischium and may be transported together at a different rate than they would be as separate elements. Pubic and pectoral girdle bones are the most commonly found bones in the HR Bonebed, followed closely by major long bones (Fig 10). The number of recovered bones decreases the more distal they are in the skeleton. It appears that, in general, axial and distal elements are under-represented, and long bones are over-represented. Ribs are well-represented, but one rib may be counted more than once if two fragments of the same rib are counted as two elements.

**Table 4. Bone surface modifications and fracturing at the four largest HR quarries.** Weathering, abrasion, and fracturing categories are adapted from Ryan et al. [25]. Fracturing data totals less than 100% due to a few bones fitting into unlisted categories.

| Feature | Stage/style | N | S | SE | T | Total | Percent |
|---|---|---|---|---|---|---|---|
| | | | | Count | | | |
| Weathering | 0 | 3324 | 2184 | 1528 | 466 | 7502 | 98.5 |
| | 1 | 36 | 23 | 31 | 4 | 94 | 1.2 |
| | 2 | 9 | 1 | 2 | 1 | 13 | 0.2 |
| | 3 | 3 | 0 | 5 | 2 | 10 | 0.1 |
| | | | | | | n = 7619 | |
| Abrasion | 0 | 3431 | 2236 | 1589 | 490 | 7746 | 97.0 |
| | 1 | 62 | 53 | 35 | 23 | 173 | 2.1 |
| | 2 | 32 | 11 | 9 | 2 | 54 | 0.7 |
| | 3 | 4 | 4 | 3 | 0 | 11 | 0.1 |
| | | | | | | n = 7984 | |
| Fracturing | Longitudinal | 119 | 85 | 24 | 16 | 244 | 0.3 |
| | Transverse | 719 | 185 | 140 | 19 | 1063 | 13.0 |
| | Greenstick/spiral | 921 | 779 | 582 | 237 | 2519 | 30.8 |
| | Indeterminate | 93 | 84 | 110 | 17 | 304 | 3.7 |
| | Unfractured | 1724 | 1240 | 776 | 179 | 3919 | 48.0 |
| | | | | | | n = 8049 | |

In 1970, Sternberg [11] reported a similar observation concerning element numbers and suggested that the dorsal and proximal caudal vertebrae were held together more tightly due to ossified tendons predominating in these regions of the spinal column. This can be seen in a juvenile *Edmontosaurus* [52] and may represent a class of transport separate from long bones. But, since most reported *Edmontosaurus* bonebeds are composed of disarticulated elements [3, 6, 9, 46, 53] it is difficult to generalize. Limited preservation of axial elements is a widespread phenomenon [54]. Head, neck, hand and foot bones, and distal caudal vertebrae appear to have disarticulated most readily in this species.

In the HR Bonebed, the dentary is the second-most commonly recovered element (Table 3). It is relatively large and heavy like the major limb bones, and probably disarticulated readily. Major body parts (forelimb, lower jaw, cranium, hindlimb) with synovial joints, disarticulate rapidly (weeks to months) during decomposition [55]. If disarticulation followed decomposition, the distal bones would dissociate first. Since hadrosaurid skulls do not exhibit fusion outside the braincase [56], the braincase and closely-associated elements would stay together and act as a single element during transport. This may mirror the tightly bound sacrum and dorsal vertebrae, suggesting that decomposition took place at the same time and place.

Loss of small elements can occur through a variety of processes. Small scavengers can remove peripheral, quickly disarticulating elements, but this is unlikely for the HR Bonebed due to the low number of bones exhibiting tooth marks. Likewise, attrition attributable to weathering seems unlikely due to the low percentage of weathered elements (Table 4) and the similarity in preservation quality between large and small elements. If the carcasses experienced a bloat and float scenario, distal elements would be removed first, as seen in turtles [57], crocodiles [58], Ichthyopterygia [59], and *Serpianosaurus* [60]. This could account for the low recovery rate of distal appendage elements, but not for the low axial element recovery rate since the body core is assumed to be still intact.

## Bonebeds and taphonomy

There are a number of other dinosaur bonebeds which share a suite of characteristics (including stratigraphy, sedimentology, condition of bones, and orientation), with the HR Bonebed. These include Upper Cretaceous, monodominant hadrosaur bonebeds containing primarily disarticulated elements [45] or completely or almost completely disarticulated elements [3, 6, 44, 46, 61]. A similar biotic or abiotic mechanism may be at work in all these examples [62]. Biotic factors such as carnivory appear to be rare in the literature, but teeth marks have been reported to be common in some *Edmontosaurus* assemblages [46]. Alternatively, the relative decomposition and disarticulation of carcasses can be used to estimate the duration of surface exposure [30]. The uniformity of pristine bone preservation indicates limited exposure. However, extremely limited articulation in the bonebed suggests that almost complete disarticulation prior to transport and burial occurred. The presence of concreted bones in the deposit is consistent with the burial of bones having biogenic fluids still associated and occurring shortly after disarticulation [35]. The lack of juveniles and uniformity of preservation states supports mass mortality of an older, age-segregated herd such as described by Ullmann [9].

A multimodal orientation of long bones also appears to be common in other dinosaur bonebeds. For example, a weak bimodal distribution NW to SE and SW to NE is seen in bonebeds BB030 and BB091 from the Dinosaur National Park region in Canada [1, 63], the Liscomb Bonebed in Alaska [6], the Dalton Wells Bonebed in Utah [45, 64], and the Blagoveschensk Bonebed in Russia [44]. Both Phillips [65] and Gangloff and Fiorillo [6] conclude that long bones were oriented either parallel or perpendicular to the prevailing current. Weeks [13] found a general southeasterly orientation on crossbed dip directions in the region surrounding the HR quarries. This mirrors the paleocurrent direction for the Lance Formation in the Powder River Basin [14] and is reflected in the bones from Teague Quarry when taken alone. However, the small number of elements recovered in this quarry may introduce a sampling bias since there is no significant orientation to bones within the HR Bonebed as a whole. As more area is excavated, variations in element orientation may lead us to better understand the dynamics of this debris flow.

The rich diversity of taxa found in the HR Bonebed mirrors that found in the Dinosaur Park Formation [25] but it does not have as high a percentage of theropod elements and differs with respect to the primary monodominant species. At the Hanson Ranch, the ratio of theropod elements to those of *Edmontosaurus* is slightly less than 4%, giving a predator to prey ratio of about 1:20. The ratios in other *Edmontosaurus* bonebeds range from about 1:50 in the Liscomb Bonebed [6] and 1:10 based on NISP in the Dalton Wells Bonebed [49] to an approximately 1:6 ratio based on NISP in the Danek Bonebed [46]. Considering that trophic level energetics are about 1:10, it appears that predators are under-represented in the HR Bonebed. However, Bakker [66] suggested that a 3% representation for dinosaur predators within a fossil assemblage is a likely realistic figure.

Determining an age profile of dinosaurs is difficult, however progress is being made [67]. If taphonomic processes, specifically post-depositional transport during a catastrophic event, have not altered the size-frequency distribution, bone length categories should represent animal sizes of the original population [68]. Histological criteria help clarify growth rate, but the lines of arrested growth (LAGs) are still inconclusive in determining exact yearly intervals of bone growth [69, 70– 71].

Bonebeds which are spread over a broad geographical area may likely represent catastrophic events enveloping a standing population, enabling characterization of age segregation if a species employs herding as a life strategy. A wide range of lengths have been reported for long bones in hadrosaur bonebeds. Brinkman et al. [68] found three distinct age clusters

compiled from many quarries in the Dinosaur Park Formation. Long bones found at the Standing Rock Hadrosaur Site, South Dakota [9] and the Danek Bonebed in Alberta [46] also represented different age classifications but there were no very small sizes and the most common categories were "subadult" and "adult" based on Horner et al. [42]. Recently, Wosik [71] combined size-frequency distributions and long bone histology in elements of *Edmontosaurus* from the Dinosaur Park Formation. He found four peaks in size that correlated with osteohistological analyses, suggesting that full adult body size was attained in 5 years.

Other bonebeds are limited to a much narrower range of length for long bones. Varricchio and Horner [72] found that several monodominant bonebeds from the Upper Cretaceous Two Medicine Formation consist of a single subadult size class of hadrosaurs 3–3.5 m long. They interpreted these animals as belonging to a first-year age class, suggesting that juvenile growth rates were as rapid as large ungulates today. The Sun River Bonebed infers primarily "late juveniles" with no adult material [51]. Similarly, long bones from the Prince Creek formation in Alaska were best described as belonging to "late juveniles", suggesting the possibility of age-segregation [6]. This inference has been reinforced by the discovery of bonebeds with long bones limited to 50–100% of maximum adult length, falling into the categories of "subadult" and "adult" [44]. These same limits are mirrored in the HR Bonebed which contains largely "subadult" and "adult" individuals.

Bone size is not the only method of determining age segregation. Histological data shows that LAGs are rare in bones of individuals less than 3.5 m in length (approximately 50% adult length) [42]. Complicating these criteria are the challenges of sexual dimorphism and the taxonomic descriptions of species based on possible ontogenetic differences. Variations in adult size may therefore potentially reflect individual or gender variation and not relative age [68, 73].

Taphonomic variability can also impact bonebed diversity. If a thanatocoenosis formed just prior to the reproductive season, it would likely not contain the bones from young animals [44]. In the Blagoveschensk Dinosaur Locality (BDL) bonebed, very small elements from older individuals were preserved, leading the authors to conclude that the absence of elements from young individuals is representative of the living population at that time [44]. Keeping in mind individual and gender size differences in a typical group of animals, and the rapid first-year growth postulated by Varicchio and Horner [72], we likewise hypothesize that the HR bonebed is representative of an *Edmontosaurous* herd just prior to seasonal breeding. We have found small adult bones of many different species, but not from *Edmontosaurus*. If histological and other aging techniques were to demonstrate that the smallest individuals are older than 1 year, it is likely that younger individuals remained separated at some other place, possibly in juvenile groups [6].

Bone modifications help us understand the taphonomic changes that occurred after death. Of the 7,619 elements examined for weathering, the vast majority (98.5%) fall within stage 0 (Table 3). Behrensmeyer [24] suggested that exposed bones develop Stage 1 cracking within a few months, therefore, bones from the HR bonebed were not subaerially exposed very long prior to burial. Limited skeletal element abrasion suggests that these elements were not exposed to protracted transport because either: a) they were not moved very far, or; b) that they were light and did not forcefully impact the surrounding debris flow elements or surfaces over which they travelled. Behrensmeyer [74] showed that in natural rivers, unweathered bones can travel many kilometers without significant rounding. For the HR Bonebed, a more plausible possibility is that these elements were transported in a viscous matrix, so they would not be subject to the bumping or abrasion that would round edges. The total distance travelled might be considerably farther than if in a more fluid matrix.

Of those bones exhibiting fracturing and breakage, the large percentage of spiral/oblique fractures elicits three possible explanations, all of which would need to occur fairly rapidly postmortem. There may have been significant scavenging, but less than 4% of recovered skeletal elements exhibit distinct tooth marks, limiting this possibility as an explanation for the common fracturing of skeletal elements. Secondly, one might expect some breakage could occur during hydraulic transport. However, a debris flow over a silty/sandy substrate would most likely not result in heavy abrasion [63], which could lead to breakage. Behrensmeyer [24, 74] reported that no breakage occurred when fresh large mammal bones were transported over several kilometers in natural rivers. The third possibility, trampling, may be the most likely cause of spiral/greenstick breakage in these bones. This type of breakage could easily have happened with minimal abrasion occurring if the substrate was fine-grained [75,76].

Eberth and Getty [63] suggest that there is no necessary link between scavenging and trampling. However, since such a large percentage of intact and spirally fractured elements occur in the HR Bonebed, it is possible that fracturing was caused by large dinosaurs scavenging the carcasses as indicated by the frequent association of shed carnivore teeth. This correlates well with the taphonomy of hadrosaurs found in the Princess Bonebed at Dinosaur National Park [1].

The pristine condition of these bones and the uniformity across all quarries suggests that these specimens originated from a single catastrophic event, as opposed to an attritional assemblage over time [40, 75]. They were probably not exposed for more than a short period of time and thus exposed to minimal pre-depositional weathering.

## Depositional model

Depositional environments that have been ascribed to the Lance include meandering and braided streams [77] in western Wyoming, and marginal marine and fluvio-deltaic [78], floodplain [79], or lacustrine [17] settings in central Wyoming. To the south, the Lance is attributed to fluvio-deltaic processes [18]. As a whole, the Lance is generally regarded as being of fluvial origin [80]. However, the Lance Formation exhibits a diversity of sedimentary characteristics across the extent of its outcrops [13, 80]. Much of the exposed stratigraphy in the study area on the eastern limb of the Powder River syncline consists of from one- to five-meter-thick beds of white, immature, fine grained quartz sandstone, alternating with meter scale gray to black, poorly indurated shaly siltstone to claystone. The sandstone bedding, where preserved, displays large and medium scale tabular and trough crossbedding. The sandstones often show severe disruption of bedding planes, apparently caused by soft sediment deformation during potential seismic disturbances (Fig 2; [81]). Where this disturbance is evident, underlying beds of sandstone may be unaffected. This suggests the affected sand units were subjected to a massive seismic event before dewatering could occur. This is more readily accommodated in an aquatic setting where dewatering may be delayed for up to six months [82]. Because these features are present at numerous horizons within the Lance, the seismic disturbance could not be directly attributable to a single event such as, for instance, an asteroid impact, and the explanation must be sought in other tectonic processes such as mountain building, perhaps associated with the Laramide orogeny [80]. The transport direction based on paleocurrents in the sandstones indicates the source area (and possibly the paleoslope) was from the west and northwest [14, 83].

The bones in the HR Bonebed occur everywhere as a normally graded sequence within the shaly mudstone. The sediment surrounding the bones is often a claystone with a particle size many orders of magnitude smaller than the largest bones encased in it. This lack of hydraulic equivalence is typical of bone assemblages [48]. Conventional descriptions suggest deposition

in a fluvial environment [6, 9, 51], but we cannot eliminate the possibility of a subaqueous debris flow [13, 84] perhaps even as part of a shallow delta [85]. Other authors have attributed similar Cretaceous bonebeds to debris flows of various types [9, 45, 64, 75, 86,87]. While this bonebed shares some of the features identified by Britt et al. [45], it also has significant differences. It is a tabular deposit, probably broadly lenticular, and elements are generally concentrated near the bottom of the unit. This deposit differs from the criteria of Britt et al. [45] in being consistently free of pebbles or cobbles and being generally well graded with respect to bones. It also does not rest on a paleosol, and most elements are from adult animals and are not broken prior to burial. Aquatic vertebrates are also relatively common. The upper boundary of the bonebed unit is in sharp and planar contact with the overlying fine-grained sandstone. Such a contact is more readily explained in a deeper water environment than in a fluvial setting, where a change from claystone deposition to sandstone would be more likely to produce an erosive contact.

In our opinion, the best explanation for the transport and deposition of the bones and the containing sediment is that they were carried together as part of a seismically activated subaqueous debris flow possibly originating from a terrestrial source an unknown distance to the west. Because the bones exhibit very little transport-induced abrasion, the flow must have been matrix-supported [88].

The scenario surrounding the demise of these animals and the formation of this deposit appears to be very similar to that depicted for other Upper Cretaceous *Edmontosaurus* bonebeds [3, 9, 45,46, 51]. A large herd of *Edmontosaurus* died, either by drowning, or by some other mass catastrophic process. The carcasses may have bloated and floated for a period of time, but eventually their remains became stranded and accumulated in a nearshore environment. Here, they were subjected to scavenging by various carnivores, including tyrannosaurids, dromaeosaurids, and troodontids while decay occurred. If the habits of modern predators/scavengers were operative in the Cretaceous, the food target of choice would be the ribcage and the soft organs inside [89]. Since dinosaurs had a more extensive ribcage than modern mammals, accessing the organs may have entailed biting through the ribs. This could explain why the ribs, which are generally resistant to transport breakage, are often found in pieces with edges that exhibit shearing.

Aerial exposure was limited to a period of time long enough (weeks to months) to result in decay-mediated disarticulation of the skeletons but was not long enough (months to years) for weathering to affect the bones. At this point, it appears probable that a significant fluvial flood or a regional seismic event initiated a significant influx of fine-grained sediment which led to suspension of the bones within a cohesive matrix, enveloping the mass of disarticulating skeletons and transporting the mixture basinward within a debris flow. This resulted in the formation of the normally graded bonebed in a matrix of clay and silt.

## Comparison of bonebeds

Over the past few decades, authors describing hadrosaur bonebeds have started including data on element representation [1, 3,4, 6, 9, 51]. There is currently no formal protocol, as some authors have grouped elements into various categories such as hindlimb long bones, forelimb bones [1], specific elements as defined in Voorhies groups [3], or more complete lists of individual bones [6, 9, 51]. Those with comparable data are included in Table 4. Correlations with HR Bonebed elements are presented in Fig 12 and Table 5.

There is a remarkably similar skeletal representation among all of the assemblages. Correlation between our percentages and those of Christians [53] (p = < 0.001), Gangloff and Fiorillo [5] (p = <0.001), Scherzer and Varricchio [51] (p = < 0.001), and Ullmann et al. [9]

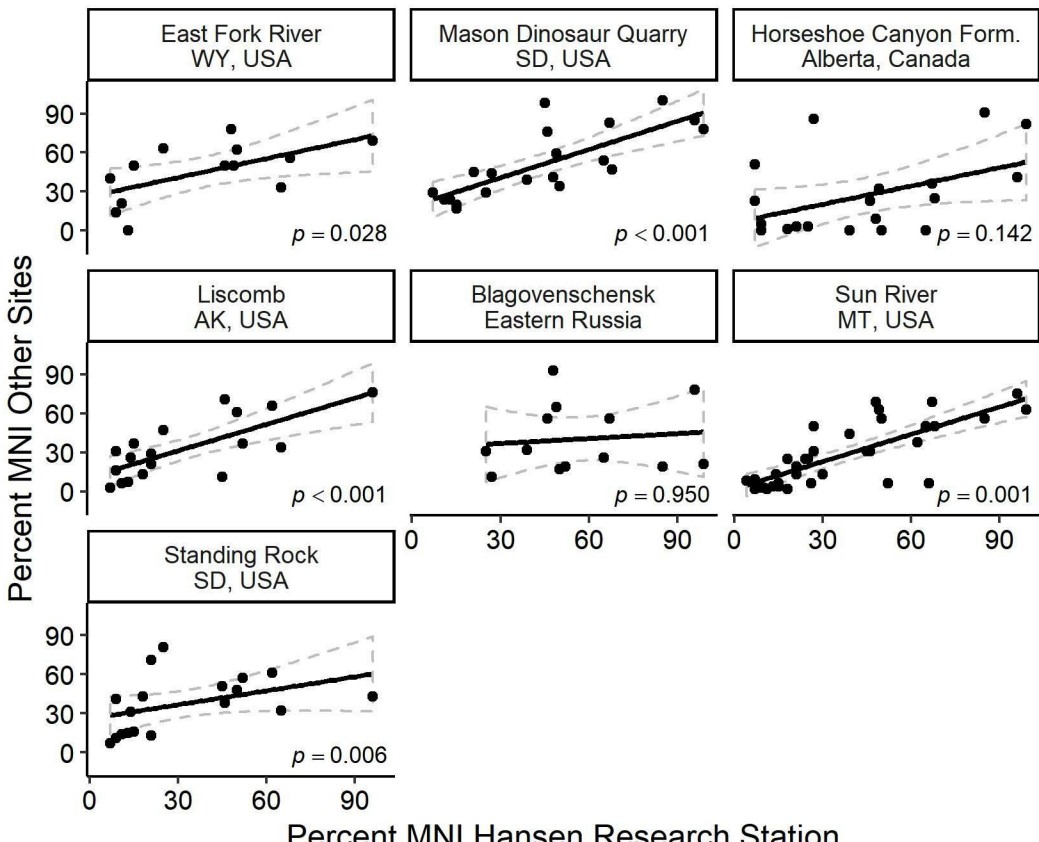

**Fig 12. Correlations of percent MNI among published hadrosaurid bonebeds.** The closer the individual dots are to the line, the more significant the correlation is. The gray dotted lines indicate the 95% confidence intervals around each black best-fit line. East Fork River [90], Mason Dinosaur Quarry [10], Liscomb [6], Blagovenschensk [44], Sun River [51], Standing Rock [9].

(p = 0.006), are all significant. There is even a significant correlation with fluvial assemblages of bones of modern vertebrates as reported by Aslan and Behrensmeyer [90] (p = 0.012). The large number of bone fragments reported by Evans et al. [3] may explain why that bonebed in the Horseshoe Canyon Formation (Table 5) is not significantly correlated with the HR Bonebed (p = 0.142). These bonebeds are spread over a wide geographical area–Alaska, Alberta, Montana, South Dakota, and Wyoming. They also include ontogenically different ages with both adult/subadult ([9], this paper) and late juvenile animals [6, 51] as defined by Horner et al. [42]. The same taphonomic factors may be acting in all these bonebeds, since they share high percentages of large proximal limb bones (and the dentary) and low representation of small bones, the dorsal and sacral vertebrae, and elements of the chondrocranium. Even in a distant bonebed [86], dentaries and humeri are the most abundant skeletal elements recovered. As more data become available, we hope the robustness of these comparisons can increase.

In our opinion, the most probable cause of HR Bonebed element distribution is that of hydraulic transport, deposition, and winnowing. Assuming most elements were disarticulated by decomposition prior to burial, a subsequent hydraulic event carried the bones some distance. Those elements that were denser (e.g., braincase, sacrum) were not moved as far as those that were smaller or more buoyant. Still-articulated elements (dorsal and caudal

**Table 5. Element comparisons among different bonebeds.** All except East Fork River are dinosaurs. Percent of total possible elements based on each bonebed MNI are listed. Percent of recovered extant bones are listed for East Fork River. Data for the other sites were derived from the following references: Standing Rock [9], Sun River [51], Blagovenschensk[44], Liscomb[6], Horseshoe Canyon [3], Mason Dinosaur Quarry [10], East Fork River [90]. Abbreviations: MNI, minimum number of individuals; Vert., vertebra.

| Region | Element | Hanson Ranch | Standing Rock | Liscomb | Sun River | Horseshoe Canyon | Blagovenschensk | Mason | East Fork River |
|---|---|---|---|---|---|---|---|---|---|
| | #Elements | 8460 | 4702 | 2769 | 537 | 1612 | 1500 | 3000 | 311 |
| | Basisphenoid | 15 | 16 | 37 | | | | | |
| | Exoccipital | 21 | 13 | 21 | 19 | | | | |
| | Jugal | 52 | 57 | 37 | 6 | | 19 | | |
| | Maxilla | 45 | 51 | 11 | 31 | | | 98 | |
| | Nasal | 66 | | | 6 | | | | |
| | Parietal | 9 | 11 | 16 | | | | | |
| | Premaxilla | 24 | | | 25 | | | | |
| | Quadrate | 62 | 61 | 66 | 38 | | | | |
| | Squamosal | 26 | | | 6 | | | | |
| | Frontal | 18 | 43 | 13 | 25 | | | | |
| | Postfrontal | 30 | | | 13 | | | | |
| | Postorbital | 14 | 31 | 26 | 13 | | | | |
| Mandible | Dentary | 96 | 43 | 76 | 75 | 41 | 78 | 85 | 69 |
| Axial | Vert., caud | 9 | 41 | 31 | 3 | 5 | | | |
| | Vert., cerv | 18 | | | 2 | 1 | | | |
| | Vert., dors | 7 | 7 | 3 | 9 | 51 | | | 40 |
| | Chevron | 15 | | | 4 | | | 20 | |
| | Rib | 68 | | | 50 | 25 | | 47 | 6 |
| | Rib, cervical | 10 | | | 3 | | | | |
| | Clavicle | 27 | | | 31 | | | | |
| | Coracoid | 7 | | | | 23 | | 29 | |
| | Humerus | 48 | | | 69 | 9 | 93 | 41 | 78 |
| | Metacarpal | 21 | 71 | 29 | 13 | 3 | | 45 | |
| | Phalanx, manus | 11 | 14 | 6 | 2 | | | 24 | 21 |
| | Radius | 50 | 48 | 61 | 56 | 0 | 17 | 34 | 62 |
| | Scapula | 65 | 32 | 34 | 50 | 0 | 26 | 54 | 33 |
| | Ulna | 39 | | | 44 | 0 | 32 | 39 | |
| | Ungual, manus | 7 | | | 2 | | | | |
| Pelvic | Astragalus | 15 | | | 6 | | | 17 | 50 |
| | Femur | 49 | | | 63 | 32 | 65 | 59 | 50 |
| | Fibula | 67 | | | 69 | 36 | 56 | 83 | |
| | Ilium | 27 | | | 50 | 86 | 11 | 44 | |
| | Ischium | 85 | | | 56 | 91 | 19 | 100 | |
| | Metatarsal | 25 | 81 | 47 | 25 | 3 | 31 | 29 | 63 |
| | Phalanx, pes | 13 | 15 | 7 | 4 | | | 24 | 0 |
| | Pubis | 99 | | | 63 | 82 | 21 | 78 | |
| | Sacrum | 9 | | | | 0 | | | 14 |
| | Tibia | 46 | 38 | 71 | 31 | 23 | 56 | 76 | 50 |
| | Ungual, pes | 4 | | | 8 | | | | |
| MNI | | 61 | 44 | 36 | 8 | 11 | 36 | 20 | Varies |

vertebrae) may have also resisted transport and acted as a single denser unit. This seems reasonable since we find the remains are normally graded. The erosion around our sites indicates that we are currently excavating what is likely the central portion of the HR Bonebed. We

propose that few of the denser elements reached our position in the bonebed during transport, and the smaller, less dense elements may have been carried beyond our current quarries.

However, we cannot rule out the opposite possibility shown for mammalian elements, that some articulated skeletal elements are transported farther than when they are isolated [90,91]. Additionally, elements that have flesh (specific gravity ~1) still attached would float more easily than bone alone (specific gravity ~1.3) [92]. Larger articulated elements would be more likely to retain tissues and float, giving them a much different hydraulic character than isolated elements.

## Conclusions

The Hanson Ranch Bonebed, exposed in five main and three exploratory quarries, is a mono-dominant hadrosaurid bonebed, consisting of sub-adult to adult *Edmontosaurus annectens*. Core body elements (vertebrae, sacra, braincases) and distal appendage elements (phalanges, carpals, predentaries) are recovered in low numbers. Proximal appendicular elements and dentaries are the most abundant bones in the assemblage. This suggests element transport and distribution based on size or relative weight. Comparisons with similar bonebeds across North America show a significant correlation in skeletal representation among sites, which implies they may share similar taphonomic histories. The most parsimonious scenario for the HR Bonebed is that of a large number of individuals dying in a single catastrophic event, followed by decomposition over weeks to months, because HR bones are generally in pristine condition. Shed theropod teeth suggest that scavenging occurred, but the large number of carcasses may be why there are very few tooth traces on bones. A single subaqueous debris flow then moved the disarticulated elements far enough to permit vertical sorting of elements before coming to rest. The primarily silt and clay matrix must have been viscous enough and transport brief enough to provide protection from abrasion, but liquid enough for sorting and grading to have taken place. This sub-aqueous debris flow transported bones and the surrounding sediment into deeper water. According to Moore [28], a large majority (13 of 16) of other Upper Cretaceous dinosaur bonebeds are recognized as being deposited in a catastrophic setting, and this bonebed appears to follow that trend.

## Acknowledgments

We thank the Hanson Ranch owners for their helpfulness in facilitating this research and Aaron Corbit for statistics and figure development. We would also like to thank Benjamin Streit for his invaluable help in the lab.

## Author Contributions

**Conceptualization:** Keith Snyder, Matthew McLain, Arthur Chadwick.

**Data curation:** Jared Wood, Arthur Chadwick.

**Formal analysis:** Keith Snyder, Matthew McLain, Arthur Chadwick.

**Funding acquisition:** Jared Wood.

**Investigation:** Keith Snyder, Matthew McLain, Jared Wood, Arthur Chadwick.

**Methodology:** Keith Snyder, Matthew McLain, Arthur Chadwick.

**Project administration:** Jared Wood, Arthur Chadwick.

**Resources:** Jared Wood, Arthur Chadwick.

**Software:** Arthur Chadwick.

**Supervision:** Keith Snyder, Arthur Chadwick.

**Writing – original draft:** Keith Snyder, Matthew McLain, Arthur Chadwick.

**Writing – review & editing:** Keith Snyder, Jared Wood.

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
