## [Decision Letter · Decision Letter 0]

12 Feb 2020

PONE-D-20-01504

Over 13,000 elements from a single bonebed help elucidate disarticulation and transport of an Edmontosaurus thanatocoenosis.

PLOS ONE

Dear Dr. Wood,

Thank you for submitting your manuscript to PLOS ONE. After careful consideration, we feel that it has merit but does not fully meet PLOS ONE’s publication criteria as it currently stands. Therefore, we invite you to submit a revised version of the manuscript that addresses the points raised during the review process.

We would appreciate receiving your revised manuscript by Mar 28 2020 11:59PM. To enhance the reproducibility of your results, we recommend that if applicable you deposit your laboratory protocols in protocols.io, where a protocol can be assigned its own identifier (DOI) such that it can be cited independently in the future. For instructions see: http://journals.plos.org/plosone/s/submission-guidelines#loc-laboratory-protocols

We look forward to receiving your revised manuscript.

Kind regards,

Jun Liu

Academic Editor

PLOS ONE

Additional Editor Comments (if provided):

It is a lot work and worth to publish. But see comments by the reviewer, especially five points proposed by one reviewer. Please revise the ms after the sugestions. Do not forget the attached word file.

Journal Requirements:

2. We note that Figures 1 and 4 in your submission contain map and satellite images which may be copyrighted. All PLOS content is published under the Creative Commons Attribution License (CC BY 4.0), which means that the manuscript, images, and Supporting Information files will be freely available online, and any third party is permitted to access, download, copy, distribute, and use these materials in any way, even commercially, with proper attribution. For these reasons, we cannot publish previously copyrighted maps or satellite images created using proprietary data, such as Google software (Google Maps, Street View, and Earth). For more information, see our copyright guidelines: http://journals.plos.org/plosone/s/licenses-and-copyright.

You may seek permission from the original copyright holder of Figures 1 and 4 to publish the content specifically under the CC BY 4.0 license. 

If you are unable to obtain permission from the original copyright holder to publish these figures under the CC BY 4.0 license or if the copyright holder’s requirements are incompatible with the CC BY 4.0 license, please either i) remove the figure or ii) supply a replacement figure that complies with the CC BY 4.0 license. Please check copyright information on all replacement figures and update the figure caption with source information. If applicable, please specify in the figure caption text when a figure is similar but not identical to the original image and is therefore for illustrative purposes only.

Reviewers' comments:

Reviewer's Responses to Questions

**Comments to the Author**

1. Is the manuscript technically sound, and do the data support the conclusions?

Reviewer #1: Yes

Reviewer #2: Yes

2. Has the statistical analysis been performed appropriately and rigorously? 

Reviewer #1: I Don't Know

Reviewer #2: Yes

3. Have the authors made all data underlying the findings in their manuscript fully available?

Reviewer #1: Yes

Reviewer #2: Yes

4. Is the manuscript presented in an intelligible fashion and written in standard English?

Reviewer #1: Yes

Reviewer #2: Yes

5. Review Comments to the Author

Reviewer #1: This paper has strong scientific merit, due to the large amount of data that supports its conclusions, and the careful analysis and writing. The references to other literature is very adequate, and shows strong familiarity with that literature.

My comments deal with minor revisions, and I will mention a few small mistakes. I think the statistics are OK, but I am not an expert in statistics.

Formatting and clarity:

Line 152 - clarify this sentence, perhaps - Field techniques for documenting bones and their position, using . . . .

When did the GPS system begin to be used in this research?

Line 159 - should this begin a new section - perhaps Taphonomic concepts

Line 183-186 - Perhaps move this paragraph to after line 177. It might make the logical sequence better.

Line 187-188 - Is this number of citizen scientists only working days at the quarry, or does it include lab work?

Line 211-212 - should it say "much of our research", or "all of this paper"

Reference to Table 2 - Perhaps the first reference to Table 2 should be the one in line 314 (?)

Line 339-340 - Does this comment apply only to the Stair quarry, or to all quarries?

Line 408 - Perhaps it should be stated how the 6896 elements were chosen.

Line 490 - Could there be a brief concluding statement of this section?

Line 549-551 - Would this read more clearly if this sentence came earlier, perhaps at line 528?

My comments on the figures are probably answered in the high resolution versions.

Fig. 2 and 3 need more clarity.

Figure 4 - Would the black dots show up better in a different color?

Figure 6 - It seems that this figure would use space better if the columns were closer together, with a larger font.

Minor points:

Line 29 - weathering, or breakage, instead of and

Line 54 and 692 - should it be ontogenic or ontogenetic?

Line 145 - Specimen Collection, preparation . . .

Line 157 - Each element was also photographed . . .

Line 247 - Would "are similar to" be better than "reflect"

Line 362 - italicize the species name

Line 504 - say "the same" rather than "this"

Line 565 - typo in genus name

Line 721 - clarify the phrase "skull dentaries"

Reviewer #2: The authors provide a detailed study of the depositional and taphonomic history of a vast bonebed of Edmontosaurus in eastern Wyoming. They apply both traditional and GIS methods toward documenting and studying the assemblage, and compare the HR Bonebed with other hadrosaurid bonebeds across the globe to glean insights into the paleobiology of these dinosaurs. Ultimately, the authors provide another piece to the puzzle supporting the idea of herding and age segregation of herds in hadrosaurid dinosaurs.

Overall, I find the data to be generally sound and the suite of analyses to be detailed/rigorous, and the assemblage is intriguing for its differences from other recently-published hadrosaurid bonebeds. I applaud the authors for their GIS-based 3D reconstructions of the assemblage, which are downright incredible, as are the online resources at the listed link, which sets a new standard for the documentation of bonebeds and making records of them open access.

However, the text could benefit from some significant editing and addition of citations in places, and I frequently was uncertain as to what the writing was trying to convey. A thorough pass aimed at clarification would be useful. I tried to give suggestions on potential modifications that could help when possible, which I hope the authors will consider, and pose questions in other places to note where clarification is needed. For example, there are a few linkages between taphonomic data and interpretations that sound inaccurate (e.g., lack of articulation doesn’t suggest mass mortality, but the limited ontogenetic range represented and uniform patterns of weathering and abrasion do). Similarly, data presented in a couple of the tables could use a little better clarification (e.g., “subtotal” lines in Table 3). Multiple figures (2, 3, 5, 6, 7, 8[component reordering]) could also use some editing to aid clarification.

Scientifically, in my opinion there are five main topics that either need revision or to be addressed. First, I agree the species is most likely E. annectens, but all of the data provided in support of this conclusion are circumstantial - this assignment needs to be documented by anatomical comparisons. Second, the authors should use age categories developed for Edmontosaurus by Gangloff and Fiorillo (2010) rather than those developed for ceratopsians by Ryan et al. (2001). Third, the authors largely disregard abundant isolated teeth, ossified tendons, and bone fragments in the assemblage, but these must also be studied as their preservation states will likely provide further (and potentially very useful) insights into the transport history of the assemblage (especially given their great abundances). Presuming these are also in the collection, this addition should not require too much time.

The other two topics of importance are interconnected, and deal with the interpretation of the depositional scenario and environment. I agree the assemblage looks like a debris flow deposit, with evidence such as the normal grading of bones and paucity of lightweight skeletal elements supporting this interpretation. Because of this, it seems far more likely that the common spiral breakage observed is best attributable to relatively high energy transport of the bones in the skeletally-dense debris flow, not to postmortem trampling. Though the authors suggest that hydraulic inequivalency (sensu Behrensmeyer 1975) may reflect the remains being authochthonous, this interpretation is inconsistent with all the other data – short distance transport by a quickly moving large volume of mud can create a parautochthonous end deposit like this exhibiting hydraulic inequivalency between bones and their host matrix, and this type of event is not uncommon in fluvial environments. Finally, it appears an outdated reference (their #76) may have lead the authors partly astray in interpreting the environment of burial, specifically into considering a potential marine depositional setting for the HR Bonebed. The entire Lance Formation is now widely agreed to be of fluvial origin because the marine Cannonball Formation has been definitively clarified to be Paleocene in age, and thus it is not a member of the Lance. Many workers indeed consider the Lance to be the same deposits as the Hell Creek Formation across the MT state line, deposits of which are also of fluvial origin. Thus, the debris flow hosting the HR Bonebed must have been deposited on a floodplain. The authors’ data are consistent with this, as only terrestrial and fluvial/freshwater taxa are represented and the sedimentology of the bonebed unit and surrounding strata are consistent with fluvial channel sands and floodplain mudstones. Though the comments in my review on this point are long, it is really a rather minor point that impacts only where you interpret the final site of burial to be. The primary changes will mostly be altering phrases from “marine” or “deltaic” to “aquatic” or “fluvial”, and so forth.

Please see the uploaded Word document for additional line-by-line comments and suggestions.

I recommend this manuscript for revision, specifically for Major Revision given the need for a little additional work in collections and the extent of editing to the text and figures that I think is needed. Were Moderate Revision a choice (as in some journals), I probably would have gone with that. Thank you,

P. Ullmann

6. PLOS authors have the option to publish the peer review history of their article (what does this mean?). If published, this will include your full peer review and any attached files.

Reviewer #1: No

Reviewer #2: Yes: Paul V. Ullmann

---

## [Author Response · Author response to Decision Letter 0]

30 Mar 2020

Jun Liu

Academic Editor

PlOS One

Chinese Academy of Sciences

Beijing, China 

Dear Dr. Liu, 

Re: PONE-D-20-01504

Please find attached a revised version of our manuscript entitled Over 13,000 elements from a single bonebed help elucidate disarticulation and transport of an Edmontosaurus thanatocoenosis, which we are resubmitting for further consideration in the PLOS One. We would like to thank you and the reviewers for your thoughtful comments, which helped us improve the manuscript during the revision process. Below, are point-by-point responses to each of the reviewers’ critiques. 

Sincerely,

Jared Wood

Department of Biology

Southwestern Adventist University

100 W. Hillcrest St.

Keene, TX 76028 USA

+1 580-677-1608

j.wood@swau.edu

Corrections Made to Manuscript

Reviewer 1.

Formatting and clarity:

1. Line 152 - clarify this sentence, perhaps - Field techniques for documenting bones and their position, using . . . .

 Clarified this sentence removing ambiguity.

2. When did the GPS system begin to be used in this research?

 Added sentence clarifying when GPS began to be used.

3. Line 159 - should this begin a new section - perhaps Taphonomic concepts

 The sectional title is currently “Specimen Preparation and Taphonomic Classifications. We split this title to “Specimen Collection and Preparation”, and then at the suggested break (line 159) added the title “Taphonomic Concepts and Classifications” This should make the sections more easily understood.

4. Line 183-186 - Perhaps move this paragraph to after line 177. It might make the logical sequence better.

 Moved this paragraph as suggested. 

5. Line 187-188 - Is this number of citizen scientists only working days at the quarry, or does it include lab work?

 Clarified the working days as those in the quarries only. 

6. Line 211-212 - should it say "much of our research", or "all of this paper"

 Changed to “this paper”

7. Reference to Table 2 - Perhaps the first reference to Table 2 should be the one in line 314 (?)

 Eliminated first reference to Table 2 (line237). Moved Table 2 to line 317. 

8. Line 339-340 - Does this comment apply only to the Stair quarry, or to all quarries?

 Clarified that this was only Stair quarry.

9. Line 408 - Perhaps it should be stated how the 6896 elements were chosen.

 Combining with the suggestion by Reviewer 2, we clarified how the elements were chosen,

 adding teeth, tendons, and bone fragments. 

10. Line 490 - Could there be a brief concluding statement of this section?

 Added:

 Thus, it appears that although there is a complex interplay of factors affecting the disarticulation and subsequent transport of Edmontosaurus elements, a pattern emerges showing a low recovery of axial and distal appendicular elements, and a high recovery of proximal appendicular elements. 

11. Line 549-551 - Would this read more clearly if this sentence came earlier, perhaps at line 528?

 Moved and clarified.

12. My comments on the figures are probably answered in the high resolution versions.

Fig. 2 and 3 need more clarity.

 Combining with comments by Reviewer 2, we added a line drawing for Fig. 2, and reworked 

 Fig. 3 to clarify layering and resolution. 

12. Figure 4 - Would the black dots show up better in a different color?

 Changed colors to show up better. 

13. Figure 6 - It seems that this figure would use space better if the columns were closer together, with a larger font.

 Modified as suggested.

14. Minor points:

Line 29 - weathering, or breakage, instead of and - done

Line 54 and 692 - should it be ontogenic or ontogenetic? - done

Line 145 - Specimen Collection, preparation . . . - done 

Line 157 - Each element was also photographed . . . - done

 - kept term “re-photographed”

Line 247 - Would "are similar to" be better than "reflect" - done

Line 362 - italicize the species name - done

Line 504 - say "the same" rather than "this" - done

Line 565 - typo in genus name - done

Line 721 - clarify the phrase "skull dentaries" - done

Reviewer 2. 

1. However, the text could benefit from some significant editing and addition of citations in places, and I frequently was uncertain as to what the writing was trying to convey. A thorough pass aimed at clarification would be useful. I tried to give suggestions on potential modifications that could help when possible, which I hope the authors will consider, and pose questions in other places to note where clarification is needed. For example, there are a few linkages between taphonomic data and interpretations that sound inaccurate (e.g., lack of articulation doesn’t suggest mass mortality, but the limited ontogenetic range represented and uniform patterns of weathering and abrasion do).

 We have followed this suggestion by examining the flow of the paper, moving a few sections to where they fit best, making all the suggested changes on the returned manuscript to clarify concepts and terminology, and added several citations while eliminating one. This reviewer put a great amount of time into upgrading this manuscript, and we worked to incorporate his suggested changes. 

2. Similarly, data presented in a couple of the tables could use a little better clarification (e.g., “subtotal” lines in Table 3). Multiple figures (2, 3, 5, 6, 7, 8[component reordering]) could also use some editing to aid clarification.

 These have been modified as suggested. 

3. Scientifically, in my opinion there are five main topics that either need revision or to be addressed. First, I agree the species is most likely E. annectens, but all of the data provided in support of this conclusion are circumstantial - this assignment needs to be documented by anatomical comparisons. 

We performed additional analyses using diagnostic anatomical traits to confirm that our hadrosaur fossils are indeed E. annectens. These additional analyses are described in the manuscript. 

4. Second, the authors should use age categories developed for Edmontosaurus by Gangloff and Fiorillo (2010) rather than those developed for ceratopsians by Ryan et al. (2001).

 Changed to reflect aging of Edmontosaurus as developed by Horner (2000) instead of ceratopsians. 

5. Third, the authors largely disregard abundant isolated teeth, ossified tendons, and bone fragments in the assemblage, but these must also be studied as their preservation states will likely provide further (and potentially very useful) insights into the transport history of the assemblage (especially given their great abundances). Presuming these are also in the collection, this addition should not require too much time.

 These have been incorporated into the data on the tables where applicable. 

6. The other two topics of importance are interconnected, and deal with the interpretation of the depositional scenario and environment. I agree the assemblage looks like a debris flow deposit, with evidence such as the normal grading of bones and paucity of lightweight skeletal elements supporting this interpretation. Because of this, it seems far more likely that the common spiral breakage observed is best attributable to relatively high energy transport of the bones in the skeletally-dense debris flow, not to postmortem trampling. Though the authors suggest that hydraulic inequivalency (sensu Behrensmeyer 1975) may reflect the remains being authochthonous, this interpretation is inconsistent with all the other data – short distance transport by a quickly moving large volume of mud can create a parautochthonous end deposit like this exhibiting hydraulic inequivalency between bones and their host matrix, and this type of event is not uncommon in fluvial environments. Finally, it appears an outdated reference (their #76) may have lead the authors partly astray in interpreting the environment of burial, specifically into considering a potential marine depositional setting for the HR Bonebed. The entire Lance Formation is now widely agreed to be of fluvial origin because the marine Cannonball Formation has been definitively clarified to be Paleocene in age, and thus it is not a member of the Lance. Many workers indeed consider the Lance to be the same deposits as the Hell Creek Formation across the MT state line, deposits of which are also of fluvial origin. Thus, the debris flow hosting the HR Bonebed must have been deposited on a floodplain. The authors’ data are consistent with this, as only terrestrial and fluvial/freshwater taxa are represented and the sedimentology of the bonebed unit and surrounding strata are consistent with fluvial channel sands and floodplain mudstones. Though the comments in my review on this point are long, it is really a rather minor point that impacts only where you interpret the final site of burial to be. The primary changes will mostly be altering phrases from “marine” or “deltaic” to “aquatic” or “fluvial”, and so forth.

 Removed outdated reference #76 and the text referring to it. Incorporated much more recent references and clarified the possible interpretations of this bonebed with the suggested 

 terminology. 

Editor comments.

1. Please ensure that your manuscript meets PLOS ONE's style requirements, including those for file naming. The PLOS ONEstyle templates can be found at

We have done our best to make sure our paper and figures are formatted according to PlosOne guidelines. 

2. We note that Figures 1 and 4 in your submission contain map and satellite images which may be copyrighted. 

Figures 1 and 4 were created by authors and did not use outside sources. Thus copyrighting is not an issue.

---

## [Decision Letter · Decision Letter 1]

30 Apr 2020

Over 13,000 elements from a single bonebed help elucidate disarticulation and transport of an Edmontosaurus thanatocoenosis.

PONE-D-20-01504R1

Dear Dr. Wood,

We are pleased to inform you that your manuscript has been judged scientifically suitable for publication and will be formally accepted for publication once it complies with all outstanding technical requirements.

With kind regards,

Jun Liu

Academic Editor

PLOS ONE

Additional Editor Comments (optional):

Reviewers' comments:

Reviewer's Responses to Questions

**Comments to the Author**

1. If the authors have adequately addressed your comments raised in a previous round of review and you feel that this manuscript is now acceptable for publication, you may indicate that here to bypass the “Comments to the Author” section, enter your conflict of interest statement in the “Confidential to Editor” section, and submit your "Accept" recommendation.

Reviewer #2: (No Response)

Reviewer #3: All comments have been addressed

2. Is the manuscript technically sound, and do the data support the conclusions?

Reviewer #2: Yes

Reviewer #3: Yes

3. Has the statistical analysis been performed appropriately and rigorously? 

Reviewer #2: Yes

Reviewer #3: Yes

4. Have the authors made all data underlying the findings in their manuscript fully available?

Reviewer #2: Yes

Reviewer #3: Yes

5. Is the manuscript presented in an intelligible fashion and written in standard English?

Reviewer #2: Yes

Reviewer #3: Yes

6. Review Comments to the Author

Reviewer #2: Overall, the authors have done an nice job incorporating my edits, comments, and suggestions. All of the figure edits have been incorporated, making them much clearer, and the addition of anatomical evidence for the assignment of the remains to E. annectens makes the manuscript stronger. Though I still disagree on a couple points being benefitted by citations (e.g., line 587), that’s really insignificant compared to the substantial improvements the authors have made. Please note however that I have one remaining noteworthy comment still focused on the interpretation of, and wording used to describe, the end depositional setting; it is addressed below about lines 706-708, 718, and 807. I also noticed a few minor edits or spots worth quick fixes, which are listed below:

Line 41: change “,or at the same time,” to “,or while,” to make this easier to understand.

Line 46: maybe augment the ending phrase to be: “…death, and them becoming interred in similar depositional settings through space and time.”

Line 198: still needs the naming citations for each Edmontosaurus species.

Line 207: the addition of anatomical diagnoses for the remains to E. annectens is nicely resolved.

Line 246: change “discusses” to “discussed”.

Line 336: my suggestions were incorporated nicely and simply.

Line 553: doesn’t need a comma after the parenthetical offset.

Line 566: correct citation is “Ullmann et al.”, not “Ullmann”.

Lines 706-708: I still think this subjective statement should be removed, especially since it is an inference that is either difficult to cite to any source or cannot be cited because it is open to interpretation. The stark transition (at the top of the bone-bearing horizon) to a sheet-like bed of fine-grained sandstone could easily occur in fluvial settings by widespread deposition of sands onto the floodplain during a crevasse splay event. In particular, the sedimentology (i.e., homogenous structure without crossbedding) of this sandstone as described in lines 138-139 is consistent with deposition in a crevasse splay. So, I believe the fluvial/crevasse splay idea sounds pretty parsimonious. Other observations supporting this interpretation include (1) the lack of laterally persistent bedding described in the Introduction (line 128), which is typical of fluvial deposits like in the Hell Creek Formation, whereas persistently subaqueous depositional settings are generally more laterally extensive (i.e. lakes, shoreface…), and (2) the presence of only fluvial/freshwater taxa in the assemblage (listed in Table 1), suggesting a fluvial (terrestrial) setting at the time of deposition.

Line 718: on the same line of thought as lines 706-708, I still think it would be better to describe the depositional setting in line 718 as “a fluvial/floodplain environment”. “Nearshore” is usually used for an offshore (off-the-beach) setting, which just doesn’t seem to be parsimonious given the authors’ observations. Maybe saying “a coastal plain environment” might be a good compromise between the authors’ thoughts and my comments? The use of the phrase “deeper water” in line 807 is vague enough to cover all of the possibilities discussed above, so it is a suitable word choice for that sentence (in my opinion).

Line 745: probably should add a phrase like “Literature data derived from the following references:” into the caption for Fig. 12.

Thank you for the opportunity to review this manuscript and its revision. I look forward to seeing it published in the near future.

Reviewer #3: This revised ms addressed all previous concerns, and no further comments are suggested. Dinosaur taphonomy is an interest topic, and this ms provides a great example to show how to inquire this in a scientific way.

7. PLOS authors have the option to publish the peer review history of their article (what does this mean?). If published, this will include your full peer review and any attached files.

Reviewer #2: Yes: Paul Ullmann

Reviewer #3: No

---

## [Editor Report · Acceptance letter]

7 May 2020

PONE-D-20-01504R1 

Over 13,000 elements from a single bonebed help elucidate disarticulation and transport of an *Edmontosaurus* thanatocoenosis. 

Dear Dr. Wood:

I am pleased to inform you that your manuscript has been deemed suitable for publication in PLOS ONE. Congratulations! Your manuscript is now with our production department. 

With kind regards,

on behalf of

Dr. Jun Liu 

Academic Editor

PLOS ONE